# Moment- and Power-Spectrum-Based Gaussianity Regularization for Text-to-Image Models

**Jisung Hwang**        **Jaihoon Kim**        **Minhyuk Sung**

KAIST

{4011hjs,jh27kim,mhsung}@kaist.ac.kr

## Abstract

We propose a novel regularization loss that enforces standard Gaussianity, encouraging samples to align with a standard Gaussian distribution. This facilitates a range of downstream tasks involving optimization in the latent space of text-to-image models. We treat elements of a high-dimensional sample as one-dimensional standard Gaussian variables and define a composite loss that combines moment-based regularization in the spatial domain with power spectrum-based regularization in the spectral domain. Since the expected values of moments and power spectrum distributions are analytically known, the loss promotes conformity to these properties. To ensure permutation invariance, the losses are applied to randomly permuted inputs. Notably, existing Gaussianity-based regularizations fall within our unified framework: some correspond to moment losses of specific orders, while the previous covariance-matching loss is equivalent to our spectral loss but incurs higher time complexity due to its spatial-domain computation. We showcase the application of our regularization in generative modeling for test-time reward alignment with a text-to-image model, specifically to enhance aesthetics and text alignment. Our regularization outperforms previous Gaussianity regularization, effectively prevents reward hacking and accelerates convergence.

## 1   Introduction

The Gaussian distribution plays a central role in numerous machine learning applications, widely used to model measurement noise, uncertainty, and the mean of independent samples. Its mathematical simplicity and analytical tractability have made it a default modeling choice in many contexts. In particular, within *generative modeling*, the standard Gaussian is commonly used as the latent distribution mapped to complex data distributions.

Given the pervasive presence of Gaussianity not only in machine learning but also across the broader sciences, quantifying how closely a data point follows—or deviates from—a Gaussian distribution, i.e., measuring *Gaussianity*, has become a fundamental technique [1, 32, 2, 9, 22, 31, 13, 30, 16, 10, 3]. In generative modeling in particular, measuring Gaussianity can facilitate optimization by identifying the latent Gaussian variable that best maps to a desired data point, thereby enabling more precise and controllable generation. For example, in the context of widely used text-to-image models [20, 11, 6, 23, 25], prior work has shown that Gaussianity-based regularization can improve downstream tasks such as aesthetic image generation [33] and text-image alignment [12].

In this work, we study the structural properties of the standard Gaussian distribution and propose a novel regularization loss that unifies various existing approaches under a unified theoretical framework. Leveraging the identity covariance of the standard Gaussian, we treat a high-dimensional sample as a collection of i.i.d. scalar standard Gaussian variables—that is, as $D$ samples drawn from a one-dimensional standard Gaussian. With a specific ordering, we analyze the distribution of these scalar values in both the one-dimensional *spatial* and *spectral* domains. In the spatial domain, we utilize the *moments* of the samples—the expected values of their powers—and define a moment-based

regularization loss. Notably, previously proposed losses such as norm-based [27, 12], kurtosis-based [7], and KL-divergence-based [18] regularization can be interpreted as, or shown to be asymptotically equivalent to, specific instances of this moment-based formulation.

Spatial-domain regularization alone is often insufficient in generative modeling, as it can leave residual patterns that lead to mappings to unrealistic data points (Figure 1 (c)). To address this, we further analyze the samples in the spectral domain and introduce an additional regularization loss based on the fact that the empirical power spectrum of i.i.d. Gaussian samples follows a chi-square distribution. This corresponds to fitting the sample covariance to the identity matrix in the spatial domain, aligning with prior work [33]. However, by operating in the spectral domain, we reduce the computational complexity from $\mathcal{O}(D^2)$ to $\mathcal{O}(D \log D)$, thereby eliminating the need for dimension-wise sampling.

Crucially, because both spatial and spectral regularization should hold regardless of the ordering of elements in a high-dimensional Gaussian sample, we enforce permutation invariance by applying our losses to randomly permuted versions of the input.

As applications of Gaussianity regularization, inspired by Eyring *et al.* [12], we showcase test-time reward alignments using a one-step text-to-image model [20], specifically for enhancing image aesthetics and text alignment. Reward alignment refers to the task of optimizing the latent sample of a generative model so that the resulting output maximizes a given reward function, such as aesthetic quality or textual alignment. However, directly optimizing the latent can lead to overfitting to the reward signal—known as reward hacking—which often degrades output quality (e.g., reduced image realism). Our Gaussianity regularization effectively mitigates this issue by encouraging the optimized latents to remain close to the original Gaussian prior. Across both aesthetic and text alignment tasks, our method outperforms existing regularization approaches, achieving the highest scores across all metrics while preventing reward hacking and accelerating convergence.

## 2 Related Work

### 2.1 Gaussianity Testing and Regularization

The Gaussian distribution is a fundamental component in statistics, and numerous methods have been developed to assess whether a set of samples conforms to it. Classical tests such as K-S, A-D, and CvM [1, 32, 2, 9, 22] measure discrepancies between empirical and theoretical cumulative distribution functions. Other approaches assess Gaussianity using order statistics [31] or quantile alignment [13, 30]. These methods, however, are based on non-differentiable functions and are therefore limited in their applicability to optimization-based techniques in machine learning.

There have also been differentiable approaches applied as regularization across a range of machine learning applications. A classical example is the KL divergence used in variational autoencoders (VAEs) [18] to measure the difference between the approximate posterior and a standard Gaussian prior—though this operates on distributional parameters rather than directly on samples. Chmiel *et al.* [7] introduced a kurtosis-based regularizer that penalizes heavy-tailed behavior in model weights to improve training stability. More recently, norm-based regularization [27, 12, 4] has been used to constrain latent vectors to lie on a hyperspherical shell, matching the expected norm of a unit Gaussian. However, these methods only address marginal statistics and do not account for inter-component dependencies. We propose a unified approach that incorporates these approaches as components, with the relationships detailed in Section 3.1.3.

Probability-Regularized Noise Optimization (PRNO) [33] is a notable example that aligns the empirical covariance matrix of latent samples with the identity, thereby capturing inter-component dependencies. While effective, this method incurs quadratic memory and time complexity, making it less scalable in high-dimensional settings. In our unified framework, we propose a more efficient regularization that achieves the same objective but in the spectral domain, with details provided in Section 3.2.2 and Appendix C.

### 2.2 Reward Alignment via Latent Noise Optimization

Reward alignment refers to the process of steering generative models to produce outputs that maximize a given reward function. While previous approaches using direct fine-tuning [26, 8] or reinforcement learning [5, 34, 37] have been widely studied, these methods are computationally expensive and require retraining the model for each new reward—posing significant scalability challenges.

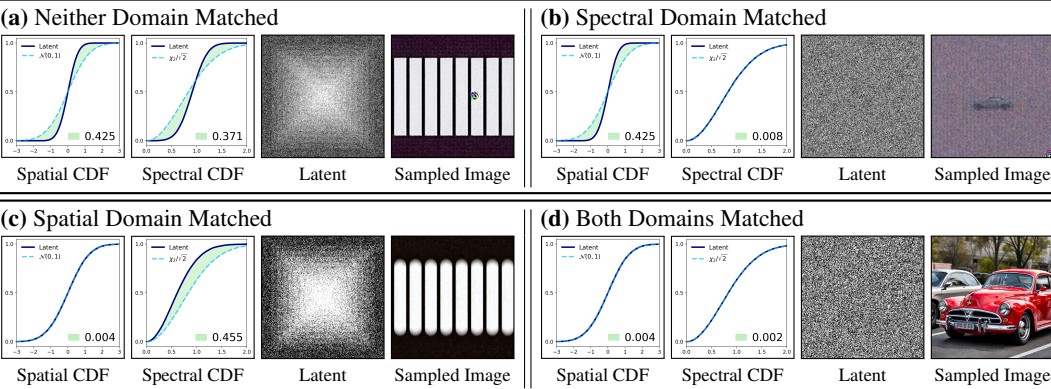

Figure 1: **Effect of Spatial and Spectral Distribution.** Each block presents the spatial and spectral CDFs, a latent visualization, and the generated image. The numeric value in each CDF plot quantifies the deviation from the ideal Gaussian distribution (lower is better). All images are generated using FLUX [20] with the prompt "A car." When both spatial and spectral properties are matched (**d**), the output is clean and realistic; mismatches in either domain lead to visible degradation in quality.

In contrast, noise optimization offers an efficient alternative by directly optimizing the initial Gaussian noise at inference time, improving the outputs of pretrained generative models without additional training. Due to its effectiveness, noise optimization has been widely applied in various domains, including images [27, 12, 4, 33], motion [17, 38], and music [29].

Recently, ReNO [12] demonstrated that noise optimization can be effectively applied to one-step generative models, offering superior reward alignment compared to multi-step models thanks to the much clearer expectation of the final output from the latent sample. However, ReNO builds on a previous Gaussianity regularization method [27], which covers only part of our unified framework. As a result, the optimized latent samples often deviate from the standard Gaussian prior, leading to degraded image quality. We show that our unified regularization more effectively preserves Gaussianity, maintaining high image quality while maximizing the reward.

## 3 Regularization for Standard Gaussianity

We propose a regularization term that captures the structural properties of the standard Gaussian distribution and unifies several existing methods. Let $\mathbf{x} = (x_1, \ldots, x_D) \in \mathbb{R}^D$ be a latent vector with i.i.d. components, $x_i \sim \mathcal{N}(0, 1)$. We specifically view $\mathbf{x}$ as an ordered sequence of samples, where any random permutation should yield a statistically equivalent vector. Thus, $\mathbf{x}$ must exhibit both the correct marginal distribution and no inter-component dependencies. A valid regularization must enforce both aspects of i.i.d. Gaussianity.

With a specific ordering of the sample sequence, our method enforces Gaussianity in both the *spatial* and *spectral* domains. In the spatial domain, we match moments of the elements in $\mathbf{x}$, generalizing several existing methods such as KL-divergence-based [18], kurtosis-based [7], and norm-based [27, 12] regularization. In the spectral domain, we match the empirical power spectrum, leveraging the fact that the empirical power spectrum of i.i.d. standard Gaussian samples follows a chi-squared distribution.

We observe that this dual-domain regularization is especially crucial for the latent samples of *generative models*, which assume an i.i.d. standard Gaussian prior. As shown in Figure 1, enforcing Gaussianity in only one domain is insufficient—spatial and spectral properties are complementary and must both be satisfied to replicate the behavior of true Gaussian samples. By jointly regularizing both domains, our method yields more faithful latent representations and improves generative performance.

In the following subsections, we formalize the spatial and spectral regularization terms in Section 3.1 and 3.2, respectively, and present our final loss formulation in Section 3.3.

### 3.1 Regularization in Spatial Domain

In this subsection, we introduce a regularization term based on the moment properties of the standard Gaussian distribution in the spatial domain. We show that this approach unifies several existing regularization methods as special cases of moment matching in the spatial domain.

#### 3.1.1 Moment Conditions of Standard Gaussianity

To formally justify our approach, we begin by recalling a classical result that uniquely characterizes the standard Gaussian distribution via its moments.

**Theorem 1.** *Let $X$ be a real-valued random variable. Suppose that for all integers $n \geq 0$, the $n$-th moment of $X$, defined as $\mathbb{E}[X^n]$, satisfies*

$$\mathbb{E}[X^n] = \begin{cases} 0 & \text{if } n \text{ is odd,} \\ \dfrac{(2k)!}{2^k k!} & \text{if } n = 2k \text{ is even.} \end{cases} \tag{1}$$

*Then $X$ follows the standard Gaussian distribution, i.e., $X \sim \mathcal{N}(0,1)$.*

*Proof.* We consider the Moment-Generating Function (MGF) of $X$:

$$\mathbb{E}[e^{tX}] = \mathbb{E}\left[\sum_{n=0}^{\infty} \frac{t^n X^n}{n!}\right] = \sum_{n=0}^{\infty} \frac{t^n}{n!}\mathbb{E}[X^n] = \sum_{k=0}^{\infty} \frac{t^{2k}}{(2k)!} \cdot \frac{(2k)!}{2^k k!} = \sum_{k=0}^{\infty} \frac{1}{k!}\left(\frac{t^2}{2}\right)^k = e^{t^2/2}. \tag{2}$$

Since this matches the MGF of the standard Gaussian distribution $\mathcal{N}(0,1)$, and the MGF uniquely determines the distribution, we conclude that $X \sim \mathcal{N}(0,1)$. □

#### 3.1.2 Moment-Based Regularization Loss

Motivated by Theorem 1, we define the following moment-based loss term to enforce the $n$-th moment condition of the standard Gaussian distribution:

$$\mathcal{L}_n = \left\lvert \left\lvert \frac{1}{D}\sum_{k=1}^{D} x_k^n \right\rvert^{1/n} - \mu_n^{1/n} \right\rvert, \tag{3}$$

where $\mu_n$ denotes the theoretical $n$-th moment of the standard Gaussian distribution, as specified in Theorem 1. This loss penalizes differences between the empirical $n$-th moment and its target value, encouraging each latent component to match the desired marginal distribution.

The computation of $\mathcal{L}_n$ scales linearly with the latent dimension $D$, resulting in both time and memory complexity of $\mathcal{O}(D)$. This makes the loss highly efficient and suitable for application to high-dimensional latent spaces commonly used in modern generative models.

#### 3.1.3 Connection to Existing Regularization Terms

The moment-based regularization introduced above is designed to reproduce the marginal statistical properties of a standard Gaussian distribution in the spatial domain. Several widely adopted regularization methods can be viewed as specific instances or constrained approximations of this principle. Below, we revisit three representative methods—KL divergence [18], kurtosis [7], and norm-based [27, 12] losses—and interpret them through the lens of spatial-domain moment matching.

**KL Regularization Loss [18].** The Kullback–Leibler (KL) divergence is widely used in the VAE framework to align the latent distribution with a standard Gaussian prior. Assuming that the empirical distribution of $\mathrm{x} \in \mathbb{R}^D$ is approximately Gaussian with empirical mean $\mu_{\mathrm{x}}$ and variance $\sigma_{\mathrm{x}}^2$, the KL divergence from the standard Gaussian is given by:

$$\mathcal{L}_{\mathrm{KL}}(\mathbf{x}) = \frac{1}{2}\left(\mu_{\mathrm{x}}^2 + \sigma_{\mathrm{x}}^2 - \log\sigma_{\mathrm{x}}^2 - 1\right). \tag{4}$$

This loss penalizes mismatches in the first and second moments of the latent distribution, encouraging both the empirical mean and variance to match those of $\mathcal{N}(0,1)$. As such, minimizing the moment

losses $\mathcal{L}_1$ and $\mathcal{L}_2$ effectively minimizes $\mathcal{L}_{\text{KL}}$. The KL loss thus serves as a compact surrogate for enforcing low-order moment alignment in the spatial domain.

**Kurtosis Regularization Loss [7].** Another approach focuses on matching the fourth central moment (kurtosis), which controls the tail behavior of the distribution. While this loss is often used to improve robustness against quantization noise in neural networks, it also serves as a valid constraint for Gaussianity enforcement. For a standard Gaussian distribution, the kurtosis is exactly 3. The kurtosis regularization loss penalizes deviation from this value:

$$\mathcal{L}_{\text{kurt}}(\mathbf{x}) = \left( \frac{1}{D} \sum_{i=1}^{D} \left( \frac{x_i - \mu_{\text{x}}}{\sigma_{\text{x}}} \right)^4 - 3 \right)^2, \quad \mu_{\text{x}} = \frac{1}{D} \sum_{i=1}^{D} x_i, \quad \sigma_{\text{x}}^2 = \frac{1}{D} \sum_{i=1}^{D} (x_i - \mu_{\text{x}})^2. \quad (5)$$

This loss encourages the latent distribution to match the fourth-order structure of the standard Gaussian. When normalization by empirical mean and variance is omitted, minimizing $\mathcal{L}_{\text{kurt}}$ becomes equivalent to minimizing the fourth-order moment loss $\mathcal{L}_4$. Even with normalization, jointly minimizing $\mathcal{L}_1$, $\mathcal{L}_2$, and $\mathcal{L}_4$ naturally reduces the kurtosis loss. Thus, $\mathcal{L}_{\text{kurt}}$ can be interpreted as a constrained variant of higher-order moment regularization in the spatial domain.

**Norm Regularization Loss [27, 12].** A common approach is to penalize deviations in the $\ell_2$ norm of the latent vector $\mathbf{x} \in \mathbb{R}^D$, assuming that each $x_i$ is independently drawn from a standard Gaussian distribution. In this case, the norm $\|\mathbf{x}\|_2$ follows a chi distribution $\chi_D$, whose probability density function is given by:

$$p_{\text{norm}}(r) = \frac{1}{2^{\frac{D}{2}-1} \Gamma\left(\frac{D}{2}\right)} r^{D-1} e^{-\frac{r^2}{2}}, \quad r \geq 0. \quad (6)$$

Maximizing the likelihood of this distribution leads to the norm-based loss:

$$\mathcal{L}_{\text{norm}}(\mathbf{x}) = -\log p_{\text{norm}}(\|\mathbf{x}\|_2) = \frac{\|\mathbf{x}\|_2^2}{2} - (D-1)\log\|\mathbf{x}\|_2 + c, \quad (7)$$

where $c$ is a constant independent of $\mathbf{x}$. This loss is minimized when the squared norm satisfies $\|\mathbf{x}\|_2^2 = D - 1$, which corresponds to an average squared component magnitude of $\frac{1}{D}\sum_{i=1}^{D} x_i^2 = 1 - \frac{1}{D}$. As $D \to \infty$, this converges to the second moment of the standard Gaussian. Thus, $\mathcal{L}_{\text{norm}}$ implicitly enforces the *second moment condition* in the spatial domain and is asymptotically equivalent to our moment loss $\mathcal{L}_2$.

In summary, many existing regularization terms can be reinterpreted as moment-matching strategies in the spatial domain. Our framework generalizes these ideas by explicitly formulating and unifying them through the lens of empirical moment alignment.

### 3.2 Regularization in Spectral Domain

As shown in Figure 1 (c), even if a latent vector has the correct marginal distribution in the spatial domain, spectral mismatch can degrade generative performance. This reflects a failure to fully reproduce the i.i.d. nature of the Gaussian prior. Hence, in this subsection, we propose a regularization term to enforce the spectral-domain properties of the standard Gaussian.

While the moment-based loss (Equation 3) in the previous section is invariant under permutations, spectral regularization is sensitive to the ordering of vector elements. The discrete Fourier transform (DFT) treats the latent vector as a structured signal, and structural dependencies—such as spatial correlations—appear in its frequency components. We focus on the power spectrum, defined as $P_k = \mathbb{E}[|\hat{x}_k|^2]$, which captures how signal energy is distributed across frequencies.

We show that for i.i.d. Gaussian vectors, the normalized DFT magnitudes $|\hat{x}_k|/\sqrt{D}$ follow a $\chi_2/\sqrt{2}$ distribution for most $k$. Based on this, we design a loss that aligns the empirical power spectrum with its expected distribution. We also relate our method to the covariance-based regularization in PRNO [33], highlighting the efficiency and theoretical grounding of our approach.

#### 3.2.1 Spectral Distribution of Standard Gaussianity

We begin by analyzing the statistical distribution of the square root of the empirical power spectrum, defined as the magnitude of the DFT coefficients: $|\hat{x}_k|$, where $\hat{x} = \text{DFT}(\mathbf{x})$. When the latent vector

$x \in \mathbb{R}^D$ consists of i.i.d. standard Gaussian samples, this distribution exhibits well-defined behavior that forms the theoretical basis for our spectral regularization.

**Lemma 1.** *Let $x \in \mathbb{R}^D$ be a random vector with i.i.d. elements $x_i \sim \mathcal{N}(0, 1)$, and let $\hat{x} = \mathrm{DFT}(x)$. Assume $D$ is even. Then,*

$$\frac{|\hat{x}_k|}{\sqrt{D}} \sim \begin{cases} \chi_2/\sqrt{2} & \text{if } k \notin \{0, D/2\}, \\ \chi_1 & \text{otherwise}. \end{cases} \tag{8}$$

The proof is provided in Appendix B. This lemma shows that when the latent vector $x$ consists of i.i.d. standard Gaussian samples, the magnitudes of its discrete Fourier transform (DFT) coefficients follow specific scaled chi distributions. For most frequency indices $k \notin \{0, D/2\}$, $|\hat{x}_k|/\sqrt{D} \sim \chi_2/\sqrt{2}$, while for $k = 0$ and $k = D/2$, $|\hat{x}_k|/\sqrt{D} \sim \chi_1$.

In practice, the latent dimension $D$ is large (e.g., 65,536 in FLUX [20]), so the bulk of the spectrum follows the $\chi_2/\sqrt{2}$ distribution. Since this property is preserved under permutation, we apply our spectral loss after randomly shuffling the latent vector to remove any ordering bias.

### 3.2.2 Power-Spectrum-Based Regularization Loss

As shown in Section 3.2.1, the magnitudes of the normalized DFT coefficients of i.i.d. standard Gaussian samples—which correspond to the square roots of the empirical power spectrum—approximately follow a $\chi_2/\sqrt{2}$ distribution for most frequency indices. A naïve approach, inspired by norm-based regularization [27], is to construct a spectral loss by applying the negative log-likelihood of this distribution to the empirical power spectrum. Specifically, we define the loss as follows:

$$\mathcal{L}_{\text{spectral\_nll}} = \sum_{i=1}^{D} \left[ -\log\left(\frac{2|\hat{x}_i|}{\sqrt{D}}\right) + \frac{|\hat{x}_i|^2}{D} \right], \tag{9}$$

where the probability density function of $\chi_2/\sqrt{2}$ is given by $f(r) = 2r \cdot e^{-r^2}$ for $r \geq 0$.

The $\chi_2/\sqrt{2}$ distribution inherently exhibits high variance, as illustrated in Figure 2. However, minimizing $\mathcal{L}_{\text{spectral\_nll}}$ drives all spectral components toward its peak (i.e., $r = 1/\sqrt{2}$), concentrating the spectrum at a single value. This contradicts the natural variability of the underlying distribution and fails to faithfully reproduce its spread. As a result, this loss formulation overly sharpens the empirical power spectrum distribution, ultimately distorting the marginal distribution of the latent vector and degrading its Gaussianity.

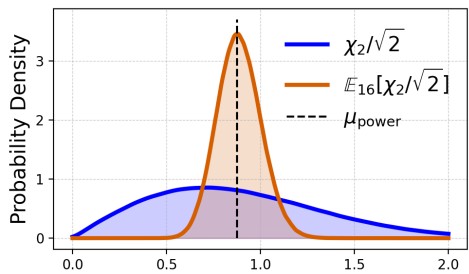

Figure 2: Distribution of $\chi_2/\sqrt{2}$ and mean of 16 independent $\chi_2/\sqrt{2}$ samples.

To mitigate this issue, we adopt an alternative strategy that preserves variance while promoting spectral alignment: instead of applying the loss to individual coefficients, we compute it over the mean of randomly sampled frequency subsets. As shown in Figure 2, the distribution of the sample mean over 16 i.i.d. $\chi_2/\sqrt{2}$ variables exhibits a bell-shaped curve with significantly reduced variance, while still reflecting the underlying distribution.

Based on this observation, we define our power spectrum regularization loss as follows:

$$\mathcal{L}_{\text{power}} = \frac{1}{|\mathcal{B}|} \sum_{B \in \mathcal{B}} \left| \frac{1}{|B|} \sum_{k \in B} \frac{|\hat{x}_k|}{\sqrt{D}} - \mu_{\text{power}} \right|, \tag{10}$$

where $\mathcal{B}$ denotes the set of batches, and $B$ represents the indices within each batch. In our experiments, we set the batch size to $|B| = 16$, and the target mean to $\mu_{\text{power}} = 0.875$, which approximates the expected value of $\chi_2/\sqrt{2}$.

This batched averaging preserves natural variation across the spectrum and avoids collapsing the distribution. It also reduces the influence of outlier frequencies at $k = 0$ and $k = D/2$, which follow different distributions. Notably, this regularization achieves the same objective as PRNO [33]—minimizing

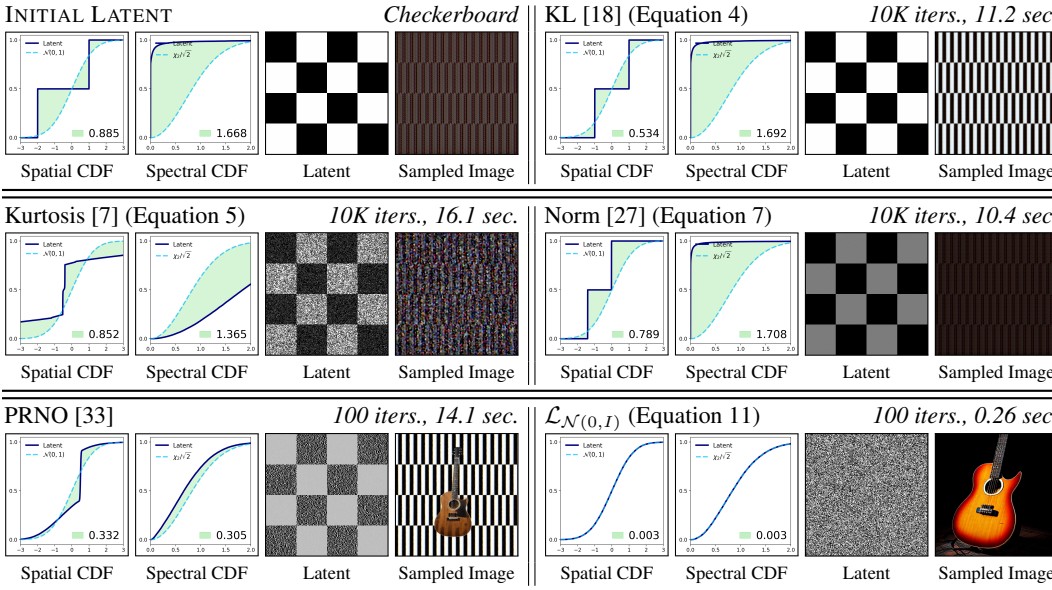

Figure 3: **Effectiveness of Regularization Losses in Guiding Latents.** A latent vector initialized with a checkerboard pattern is optimized using various regularization losses. Each block follows the format of Figure 1. Images are generated by FLUX [20] with the prompt "A guitar".

deviation from identity covariance—but enforces it in the spectral domain in a more efficient and effective manner. A detailed discussion of this connection is provided in Appendix C.

**Computational Perspective.** Calculating $\mathcal{L}_{\text{power}}$ is highly efficient due to the use of the Fast Fourier Transform (FFT) algorithm. Unlike the naive discrete Fourier transform, which has a computational complexity of $\mathcal{O}(D^2)$, the FFT reduces this to $\mathcal{O}(D \log D)$. Furthermore, FFT operations are inherently parallelizable and benefit significantly from GPU acceleration. Given that FFT is a core component in many scientific computing libraries, it is already highly optimized in most modern frameworks. As a result, we observed no significant increase in runtime compared to simpler $\mathcal{O}(D)$-based regularization methods.

### 3.3 Our Gaussianity Regularization Loss

By combining the two loss components—the moment regularization loss (Equation 3) and the power spectrum regularization loss (Equation 10)—we define our final regularization loss as:

$$\mathcal{L}_{\mathcal{N}(0,I)} = \sum_{n \in \mathcal{K}} \mathcal{L}_n + \lambda_{\text{power}} \mathcal{L}_{\text{power}}, \quad (11)$$

Table 1: Comparison of regularization methods for standard Gaussianity.

| Method | Time Complexity | Memory Complexity | Connection with Our Loss |
|---|---|---|---|
| KL [18] | $\mathcal{O}(D)$ | $\mathcal{O}(D)$ | $\mathcal{L}_1, \mathcal{L}_2$ |
| Kurtosis [7] | $\mathcal{O}(D)$ | $\mathcal{O}(D)$ | $\mathcal{L}_4$ |
| Norm [27, 12] | $\mathcal{O}(D)$ | $\mathcal{O}(D)$ | $\mathcal{L}_2$ |
| PRNO [33] | $\mathcal{O}(Dk)$ | $\mathcal{O}(Dk)$ | $\mathcal{L}_1, \mathcal{L}_{\text{power}}$ |
| **Ours** | $\mathcal{O}(D \log D)$ | $\mathcal{O}(D)$ | – |

where $\mathcal{K}$ denotes the set of moments used for matching. In our experiments, we set $\mathcal{K} = \{1, 2\}$, as enforcing the first and second moments, together with the power spectrum regularization $\mathcal{L}_{\text{power}}$, is empirically sufficient to approximate the standard Gaussian distribution. We set $\lambda_{\text{power}} = 25.0$.

### 3.4 Toy Experiment Using an Image Generative Model

We conduct a toy experiment using the image generative model FLUX [20] to evaluate how different regularization terms guide a latent vector toward a standard Gaussian distribution when optimized from a highly structured initialization. The initial latent is set to a checkerboard pattern, and the results are shown in Figure 3. Spatial-only methods—KL [18], Kurtosis [7], and Norm [27]—reduce the deviation in the spatial domain but retain strong checkerboard artifacts even after 10K optimization iterations. PRNO [33] improves alignment in both spatial and spectral domains, yet the latent still exhibits visible structure and the sampled image shows unnatural textures. In contrast, our method

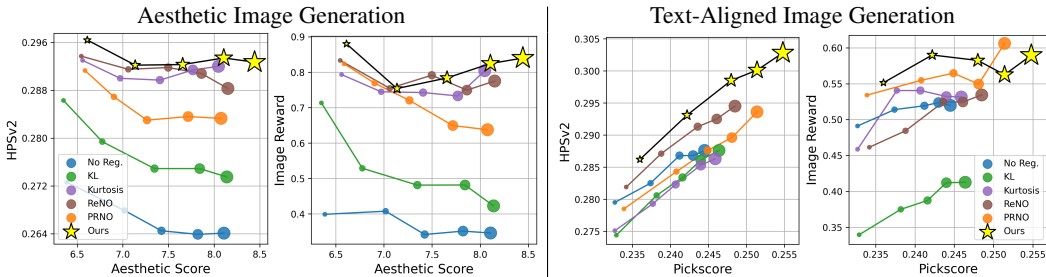

Figure 4: **Quantitative Results for Aesthetic Image Generation Text-Aligned Image Generation.** Curves show performance at 100-iteration intervals, with dot sizes indicating progress from 100 to 500 iterations. Points closer to the upper right represent better trade-offs between the given reward (x-axis) and held-out reward (y-axis). Our method reaches the highest reward with equal iterations and consistently yields better trade-offs on HPSv2 [35] and ImageReward [36].

effectively matches distributions in both domains, producing a clean, high-quality output from a noise-like latent—while requiring approximately $50\times$ less time than PRNO.

## 4 Applications: Reward Alignment in Text-to-Image Generative Models

Inspired by the previous work, ReNO [12], we present two applications of reward alignment in a one-step text-to-image generative model: aesthetic image generation and text-aligned image generation.

**Baselines.** In all experiments, we use FLUX [20] as the base generative model, which is a one-step text-to-image model. We compare our regularization method against KL [18], Kurtosis [7], ReNO (Norm) [12], and PRNO [33]. Additionally, we report two reference baselines: one without any optimization (No Opt.) and one without regularization (No Reg.).

**Implementation Details.** We initialize the latent vector from the prior distribution (a unit Gaussian) and perform optimization for 500 iterations using Nesterov momentum with a coefficient of 0.9 and gradient clipping set to 0.01. The generated images are evaluated every 100 iterations. The learning rate is set to 0.1 for aesthetic score [28] and 1.0 for PickScore [19]. We set the regularization coefficient to 2.0 for all regularization methods. The regularization gradient is normalized and scaled to match the magnitude of the reward gradient, ensuring balanced contributions during optimization. All experiments were conducted on an NVIDIA A6000 GPU with 48GB VRAM, taking approximately 2 minutes per 100 optimization iterations.

### 4.1 Aesthetic Image Generation

We use aesthetic score [28], which measures the visual appeal of an image, as the given reward—the objective used to optimize the latent. Evaluation is conducted on the animal prompts from DDPO [5]. We report the given reward along with held-out rewards—ImageReward [36] and HPSv2 [35]—which are not used during optimization and serve to assess image quality and text alignment.

**Results.** We present quantitative and qualitative results in Figure 4 and Figure 5, respectively. Figure 4 shows curves of the baseline aesthetic score plotted against the held-out rewards. The size of each dot indicates the number of optimization iterations, shown in intervals of 100 iterations.

Notably, optimizing the latent solely based on the given reward—without any regularization—leads to a phenomenon known as *reward hacking*, where the model exploits flaws in the reward function to achieve higher scores without improving, or even degrading, the actual image quality. This is evidenced by the steady decline of the blue curve in Figure 4, which indicates that the image quality—measured by the held-out rewards such as HPSv2 [35] and ImageReward [36]—deteriorates as the optimization progresses, despite the given reward increasing. The corresponding visual degradation is also apparent in the cartoon-like artifacts observed in the generated samples (column No Reg. in Figure 5). While previous works introduce regularization terms that enforce Gaussianity, these methods fall short as they fail to capture both the spatial and spectral properties of a unit Gaussian, leading to suboptimal results (see KL, Kurtosis, and ReNO columns in Figure 5). In contrast, our regularization is robust to reward hacking and consistently achieves the highest scores across all metrics—outperforming ReNO [12] and PRNO [33] at every optimization iteration.

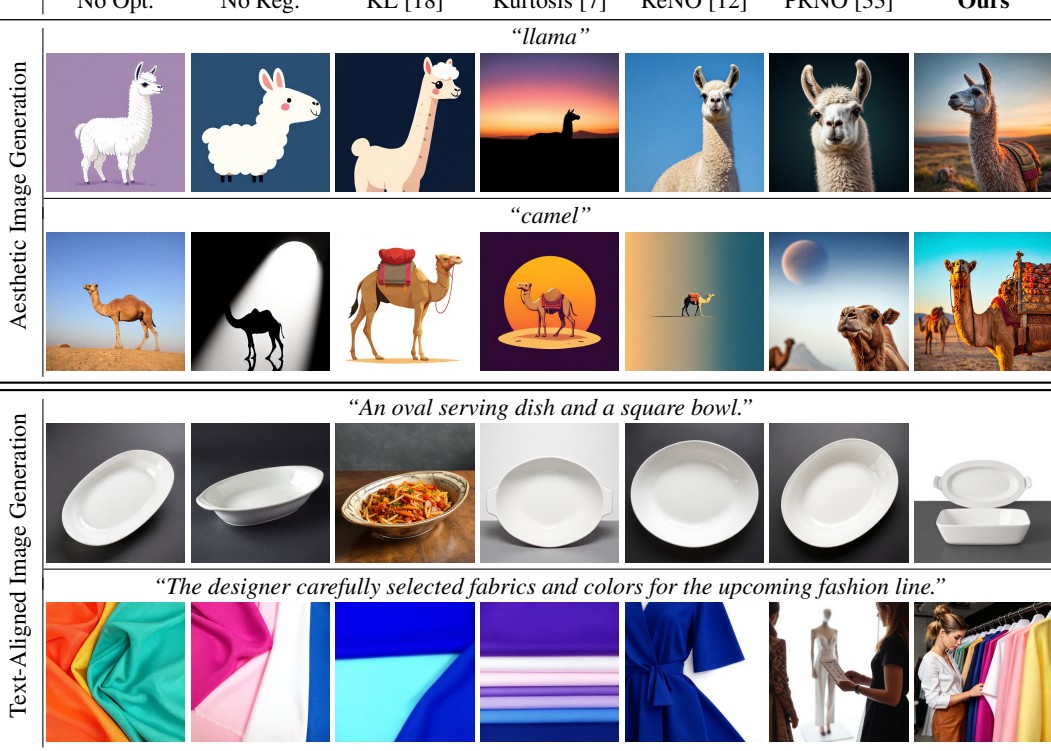

Figure 5: **Qualitative Results for Aesthetic and Text-Aligned Image Generation.** Our method generates images with higher aesthetic quality and better prompt alignment compared to baselines [18, 7, 12, 33] and not regularized optimization.

## 4.2 Text-Aligned Image Generation

For text-aligned image generation, we use PickScore [19] as the given reward, which measures both image-text alignment and perceptual image quality. This setup is a special case because the given reward is closely aligned with the objective of Gaussianity regularization which also improves the perceptual quality of the generated images in text-to-image generative models [20]. We evaluate on 60 prompts sampled from T2I-CompBench++ [15], comprising 10 prompts from each of the six categories: 3D spatial, complex, non-spatial, shape, spatial, and texture.

**Results.** Quantitative and qualitative results are presented in Figure 4 and Figure 5, respectively. As shown in Figure 4, methods incorporating Gaussianity regularization initially exhibit a strong positive scaling trend, reflecting the close alignment between the PickScore reward and the regularization objective. However, these regularization methods—including the No Reg. case—rely solely on spatial-domain constraints [18, 7, 12], and quickly plateau in performance—a limitation further evidenced by the suboptimal generation quality shown in Figure 5. By explicitly guiding latent vectors to stay close to the unit Gaussian manifold, our method promotes stable gradient flow—a property also noted in prior work [24, 14, 21]. As a result, it achieves higher rewards throughout optimization and outperforms baselines with fewer updates.

## 5 Conclusion

We introduced a unified regularization framework for enforcing standard Gaussianity. Unlike prior approaches that focus solely on marginal statistics or covariance matching, our method captures both spatial and spectral properties, improving conformity to a unit Gaussian distribution while remaining computationally efficient. We validated the effectiveness of our approach on reward alignment tasks in text-to-image generative models, including aesthetic and text-aligned image generation. Our method consistently outperforms existing Gaussianity regularization techniques by mitigating reward hacking and accelerating convergence. These results underscore the importance of structured Gaussianity enforcement and open up new directions for broader applications.

**Limitation and Societal Impacts.** While our loss effectively guides latent vectors toward Gaussianity during optimization and serves well as a regularization objective, we observe that its value alone does not reliably indicate how closely a given latent matches a true standard Gaussian distribution. That is, a low loss value does not necessarily guarantee full Gaussianity. Developing an efficient and principled metric for directly evaluating Gaussianity remains an open direction for future research. In addition, as our method builds on pretrained generative models, which may have been trained on uncurated datasets, it inherits the inherent biases and artifacts of the underlying base model. This may lead to the unintended generation of biased or undesirable visual content.

## Acknowledgements

This work was supported by the NRF of Korea (RS-2023-00209723); IITP grants (RS-2022-II220594, RS-2023-00227592, RS-2024-00399817, RS-2025-25441313, RS-2025-25443318, RS-2025-02653113); and the Technology Innovation Program (RS-2025-02317326), all funded by the Korean government (MSIT and MOTIE), as well as by the DRB-KAIST SketchTheFuture Research Center.

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

# Appendix

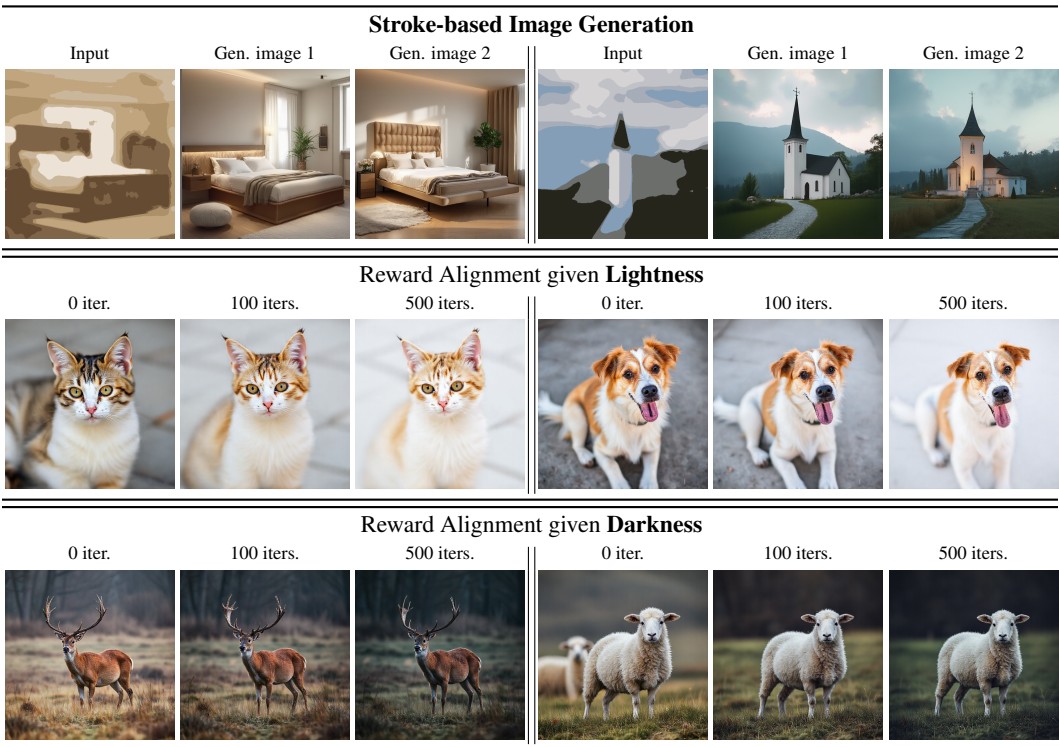

Figure A.1: **Showcase of Our Regularization Term in Other Applications.** We demonstrate the versatility of our regularization in three additional tasks: stroke-based image generation (top) and reward alignment under lightness (middle) and darkness (bottom). In the first row, images are generated from coarse stroke inputs, where our regularization helps produce semantically faithful and visually coherent outputs. In the second and third rows, we apply reward-guided optimization to adjust image lightness or darkness over time, showing smooth and stable progression from the initial sample (0 iters.) to reward-aligned outputs (500 iters.).

## A   Other Applications

To demonstrate the broader applicability of our proposed regularization, we present three additional applications, as shown in Figure A.1: stroke-based image generation (top), and reward alignment based on lightness and darkness objectives (middle and bottom, respectively).

In the stroke-based image generation task, the input is a coarse color-stroke map that encodes a rough spatial layout and scene structure. We begin by performing image inversion using the FLUX-dev [20] model to obtain an initial latent representation. We then optimize this latent by minimizing the L2 distance to the inverted latent while applying our regularization loss, which encourages the optimized latent to remain close to both the standard Gaussian manifold and the original inversion. The final image is generated from the optimized latent. This optimization step helps preserve semantic fidelity to the stroke input while enabling the generation of photorealistic images. As shown in the first row of Figure A.1, our method produces semantically aligned and visually coherent results that maintain the structure and framing of the original strokes.

In the second and third rows, we present reward-aligned image generation for lightness and darkness. For the lightness task, the reward is defined as the mean of all pixel values—encouraging brighter outputs—while for the darkness task, we use the negative of this mean to promote darker results. We apply our regularization with these rewards during latent optimization. In both cases, our method effectively guides the latent toward the desired visual attribute without sacrificing image quality.

# B  Proof for Lemma 1

**Lemma A.1.** *Let* $\mathrm{x} \in \mathbb{R}^D$ *be a random vector whose elements are i.i.d. samples from the unit Gaussian distribution, i.e.,* $x_i \sim \mathcal{N}(0, 1)$. *Let* $\hat{\mathrm{x}} = \mathrm{DFT}(\mathrm{x}) \in \mathbb{C}^D$ *denote its discrete Fourier transform. Then:*

- *For all* $k \notin \{0, D/2\}$ *(assuming* $D$ *is even), the magnitude* $|\hat{x}_k|$ *follows a scaled chi distribution with 2 degrees of freedom:*

$$\frac{|\hat{x}_k|}{\sqrt{D}} \sim \frac{\chi_2}{\sqrt{2}}. \tag{12}$$

- *For* $k = 0$ *and* $k = D/2$, *where* $\hat{x}_k \in \mathbb{R}$, *the magnitude follows a scaled chi distribution with 1 degree of freedom:*

$$\frac{|\hat{x}_k|}{\sqrt{D}} \sim \chi_1. \tag{13}$$

*Proof.* Let $\mathrm{x} = (x_0, x_1, \ldots, x_{D-1}) \in \mathbb{R}^D$ with i.i.d. $x_n \sim \mathcal{N}(0, 1)$. The DFT of x is defined as

$$\hat{x}_k = \sum_{n=0}^{D-1} x_n \cdot e^{-2\pi i k n / D}, \quad k = 0, 1, \ldots, D - 1. \tag{14}$$

This is a linear transformation with complex coefficients of unit magnitude. For $k \notin \{0, D/2\}$, $\hat{x}_k$ is complex with real and imaginary parts

$$\Re(\hat{x}_k) = \sum_{n=0}^{D-1} x_n \cos\left(\frac{2\pi k n}{D}\right), \tag{15}$$

$$\Im(\hat{x}_k) = -\sum_{n=0}^{D-1} x_n \sin\left(\frac{2\pi k n}{D}\right), \tag{16}$$

which are independent Gaussian variables with variances

$$\mathrm{Var}(\Re(\hat{x}_k)) = \sum \cos^2\left(\frac{2\pi k n}{D}\right) = \frac{D}{2}, \tag{17}$$

$$\mathrm{Var}(\Im(\hat{x}_k)) = \sum \sin^2\left(\frac{2\pi k n}{D}\right) = \frac{D}{2}. \tag{18}$$

Thus, $\Re(\hat{x}_k), \Im(\hat{x}_k) \sim \mathcal{N}(0, D/2)$, and the magnitude satisfies

$$|\hat{x}_k| = \sqrt{\Re(\hat{x}_k)^2 + \Im(\hat{x}_k)^2} \sim \sqrt{D/2} \cdot \chi_2. \tag{19}$$

which implies

$$\frac{|\hat{x}_k|}{\sqrt{D}} \sim \frac{\chi_2}{\sqrt{2}}. \tag{20}$$

For $k \in \{0, D/2\}$ (when $D$ is even), the Fourier coefficients are real:

$$\hat{x}_0 = \sum_{n=0}^{D-1} x_n, \tag{21}$$

$$\hat{x}_{D/2} = \sum_{n=0}^{D-1} x_n (-1)^n. \tag{22}$$

Each is a sum of $D$ i.i.d. standard Gaussian variables, so $\hat{x}_k \sim \mathcal{N}(0, D)$ and hence

$$|\hat{x}_k| \sim \sqrt{D} \cdot \chi_1 \quad \Rightarrow \quad \frac{|\hat{x}_k|}{\sqrt{D}} \sim \chi_1. \tag{23}$$

$\square$

# C   Connection to Probability-Regularized Noise Optimization (PRNO)

In this section, we show that the power spectrum regularization $\mathcal{L}_{\text{power}}$ (10) is a practical surrogate for the likelihood-style regularization used in Probability-Regularized Noise Optimization (PRNO) [33].

Intuitively, the regularization $\mathcal{L}_{\text{power}}$ in the spectral domain spreads energy evenly across frequencies and suppresses global offsets. Theorem A.1 formalizes this by proving that minimizing $\mathcal{L}_{\text{power}}$ drives the first and second moments to approximately 0 and 1, respectively, and renders the latent vector rotation–isotropic (i.e., with no preferred direction). Then, Theorem A.2 shows that these properties align with the objectives of PRNO by pushing the latent vector toward maximal likelihood in a broad range of cases.

We first introduce the notation and setup, and then present the two theorems with their proofs.

**Setup and notation.**   Let $D = mk = PB > 10{,}000$ be even, with the spatial block size fixed to $k = 4$ (as in the official PRNO implementation code [33]) and the frequency block size set to $B = 16$ (these constants are used in all our experiments). Let $\Pi \in \mathbb{R}^{D \times D}$ be a uniformly random permutation matrix and let $\mathbf{F} \in \mathbb{C}^{D \times D}$ denote the DFT matrix. For any $\mathrm{x} \in \mathbb{R}^D$, define the permuted signal and its DFT by

$$\hat{\mathrm{x}}(\Pi) := \mathbf{F} \, \Pi \, \mathrm{x} \in \mathbb{C}^D. \tag{24}$$

*Frequency blocks.* For $\mathcal{B}_p := \{pB, \ldots, (p+1)B - 1\}$ with selector $\mathbf{R}_{\mathcal{B}_p} \in \{0,1\}^{B \times D}$, define

$$\hat{\mathrm{x}}^{[p]}(\Pi) := \mathbf{R}_{\mathcal{B}_p} \, \hat{\mathrm{x}}(\Pi) \in \mathbb{C}^B. \tag{25}$$

Then, $\mathcal{L}_{\text{power}}$ (10) can be written as follow:

$$\mathcal{L}_{\text{power}}(\Pi \mathrm{x}) := \frac{1}{D} \sum_{p=0}^{P-1} \left| \frac{1}{\sqrt{D}} \left\| \hat{\mathrm{x}}^{[p]}(\Pi) \right\|_1 - \mu_{\text{power}} B \right|. \tag{26}$$

*Spatial blocks.* For $\mathcal{K}_i := \{ik, \ldots, (i+1)k - 1\}$ with selector $\mathbf{R}_{\mathcal{K}_i} \in \{0,1\}^{k \times D}$, define

$$\mathrm{x}^{(i)}(\Pi) := \mathbf{R}_{\mathcal{K}_i} (\Pi \, \mathrm{x}) \in \mathbb{R}^k. \tag{27}$$

Define the block mean and block covariance

$$\overline{\mathrm{x}}(\Pi) := \frac{1}{m} \sum_{i=0}^{m-1} \mathrm{x}^{(i)}(\Pi), \qquad \mathbf{S}(\Pi) := \frac{1}{m} \sum_{i=0}^{m-1} \mathrm{x}^{(i)}(\Pi) \, \mathrm{x}^{(i)}(\Pi)^\top. \tag{28}$$

Diagnostics:

$$M_1(\Pi \mathrm{x}) := \left\| \overline{\mathrm{x}}(\Pi) \right\|_2, \qquad M_2(\Pi \mathrm{x}) := \left\| \mathbf{S}(\Pi) - I_k \right\|_2. \tag{29}$$

For $M \geq 0$ define

$$p_1(M) := \min \left\{ 2 \exp\left( -\tfrac{m}{2k} M^2 \right), \, 1 \right\}, \tag{30}$$

$$\psi(M) := \left( \sqrt{1 + M} - 1 - \sqrt{k/m} \right)_+^2, \tag{31}$$

$$p_2(M) := \min \left\{ 2 \exp\left( -\tfrac{m}{4} \psi(M) \right), \, 1 \right\}. \tag{32}$$

PRNO [33] maximizes $\mathbb{E}_\Pi \left[ \log p_1(M_1(\Pi \mathrm{x})) \right]$ and $\mathbb{E}_\Pi \left[ \log p_2(M_2(\Pi \mathrm{x})) \right]$.

**Theorem A.1.** *Assume that $\mathrm{x}$ minimizes $\mathbb{E}_\Pi\big[\mathcal{L}_{\text{power}}(\Pi \mathrm{x})\big]$. Then*

$$\mu_{\mathrm{x}} = \frac{1}{D} \sum_{j=0}^{D-1} \mathrm{x}_j \approx 0, \qquad \sigma_{\mathrm{x}}^2 = \frac{\|\mathrm{x}\|_2^2}{D} - \mu_{\mathrm{x}}^2 \approx 1 \tag{33}$$

*and $\mathrm{x}$ is rotation–isotropic.*

*Proof of Theorem A.1.* Recall

$$\mathcal{L}_{\text{power}}(\Pi x) = \frac{1}{D} \sum_{p=0}^{P-1} \left| \frac{1}{\sqrt{D}} \left\| \hat{x}^{[p]}(\Pi) \right\|_1 - \mu_{\text{power}} B \right|. \tag{34}$$

To analyze the minimizer of $\mathbb{E}_\Pi\big[\mathcal{L}_{\text{power}}(\Pi x)\big]$, we begin by investigating the expectation of the magnitude of each DFT coefficient.

Let $x \in \mathbb{R}^D$ and write $x = \mu_x \mathbf{1} + \mathbf{r}$ with $\sum_{j=0}^{D-1} r_j = 0$ and $\frac{1}{D} \sum_{j=0}^{D-1} r_j^2 = \sigma_x^2$. Let $\mathbf{f}_k$ be a non-DC/non-Nyquist DFT row of $\mathbf{F}$, so $|f_j| = 1$, $\sum_{j=0}^{D-1} f_j = 0$, and $\sum_{j=0}^{D-1} |f_j|^2 = D$.

Since $\sum_j f_j = 0$,

$$\frac{1}{\sqrt{D}} \langle \mathbf{f}_k, \Pi x \rangle = \frac{1}{\sqrt{D}} \langle \mathbf{f}_k, \Pi \mathbf{r} \rangle. \tag{35}$$

Define $c_j := \overline{f_j}/\sqrt{D}$ so that $\sum_{j=0}^{D-1} c_j = 0$ and $\sum_{j=0}^{D-1} |c_j|^2 = 1$. Then

$$Y := \frac{1}{\sqrt{D}} \langle \mathbf{f}_k, \Pi \mathbf{r} \rangle = \sum_{j=0}^{D-1} c_j\, r_{\Pi(j)}. \tag{36}$$

Because $\Pi$ is uniform and the population mean of $\{r_j\}$ is 0,

$$\mathbb{E}_\Pi[Y] = 0. \tag{37}$$

Using the known covariances of sampling without replacement, $\text{Var}(r_{\Pi(j)}) = \sigma_x^2$ and $\text{Cov}(r_{\Pi(i)}, r_{\Pi(j)}) = -\frac{\sigma_x^2}{D-1}$ for $i \neq j$.

Then, we obtain

$$\text{Var}_\Pi(Y) = \sigma_x^2 \sum_{j=0}^{D-1} |c_j|^2 - \frac{\sigma_x^2}{D-1} \sum_{\substack{i,j=0 \\ i \neq j}}^{D-1} c_i\, \overline{c_j} = \left(1 + \frac{1}{D-1}\right) \sigma_x^2 = \frac{D}{D-1} \sigma_x^2. \tag{38}$$

$Y$ is a sum of many small, mean-zero terms sampled without replacement; by Hoeffding's combinatorial CLT (or Lindeberg-type arguments),

$$Y = \sum_{j=0}^{D-1} c_j\, r_{\Pi(j)} \approx \mathcal{CN}\left(0, \frac{D}{D-1} \sigma_x^2\right). \tag{39}$$

If $Z \sim \mathcal{CN}(0, \sigma^2)$, then $\mathbb{E}[|Z|] = \frac{\sqrt{\pi}}{2} \sigma$. Hence,

$$\mathbb{E}_\Pi\left[\frac{1}{\sqrt{D}} |\langle \mathbf{f}_k, \Pi x \rangle|\right] = \mathbb{E}_\Pi[|Y|] \approx \frac{\sqrt{\pi}}{2} \sqrt{\frac{D}{D-1}}\, \sigma_x. \tag{40}$$

For the DC DFT row,

$$\mathbf{f}_0 = \mathbf{1}, \qquad \frac{1}{\sqrt{D}} \langle \mathbf{f}_0, \Pi x \rangle = \frac{1}{\sqrt{D}} \langle \mathbf{1}, x \rangle = \sqrt{D}\, \mu_x. \tag{41}$$

Thus,

$$\mathbb{E}_\Pi\left[\frac{1}{\sqrt{D}} |\langle \mathbf{f}_0, \Pi x \rangle|\right] = \sqrt{D}\, |\mu_x|. \tag{42}$$

For the Nyquist DFT row $\mathbf{f}_{D/2}$ we take $f_j = (-1)^j$, hence $c_j = \overline{f_j}/\sqrt{D} = (-1)^j/\sqrt{D} \in \mathbb{R}$ and

$$\frac{1}{\sqrt{D}} \langle \mathbf{f}_{D/2}, \Pi \mathbf{r} \rangle = \sum_{j=0}^{D-1} c_j\, r_{\Pi(j)} \in \mathbb{R}. \tag{43}$$

As before,

$$\mathbb{E}_{\Pi}\left[\frac{1}{\sqrt{D}}\langle\mathbf{f}_{D/2}, \Pi\mathbf{r}\rangle\right] = 0, \qquad \mathrm{Var}_{\Pi}\left(\frac{1}{\sqrt{D}}\langle\mathbf{f}_{D/2}, \Pi\mathbf{r}\rangle\right) = \frac{D}{D-1}\sigma_{\mathrm{x}}^2. \tag{44}$$

By a finite-population (permutation) CLT in the real case,

$$\frac{1}{\sqrt{D}}\langle\mathbf{f}_{D/2}, \Pi\mathbf{r}\rangle \approx \mathcal{N}\left(0, \frac{D}{D-1}\sigma_{\mathrm{x}}^2\right), \tag{45}$$

and therefore, using $\mathbb{E}|Z| = \sqrt{2/\pi}\,\sigma$ for $Z \sim \mathcal{N}(0, \sigma^2)$,

$$\mathbb{E}_{\Pi}\left[\frac{1}{\sqrt{D}}\left|\langle\mathbf{f}_{D/2}, \Pi\mathbf{x}\rangle\right|\right] \approx \sqrt{\frac{2}{\pi}}\sqrt{\frac{D}{D-1}}\,\sigma_{\mathrm{x}}. \tag{46}$$

We analyze $Y_p := \frac{1}{\sqrt{D}}\left\|\hat{\mathbf{x}}^{[p]}(\Pi)\right\|_1 - \mu_{\mathrm{power}}B$, which constitutes $\mathcal{L}_{\mathrm{power}}(\Pi\mathbf{x})$.

The expectation over all permutations is

$$\mathbb{E}_{\Pi}[Y_p] = \sum_{k=pB}^{(p+1)B-1}\mathbb{E}_{\Pi}\left[\frac{1}{\sqrt{D}}\left|\langle\mathbf{f}_k, \Pi\mathbf{x}\rangle\right|\right] - \mu_{\mathrm{power}}B \tag{47}$$

*DC–including block ($p = 0$).*

$$\mathbb{E}_{\Pi}[Y_0] = (B-1)\frac{\sqrt{\pi}}{2}\sqrt{\frac{D}{D-1}}\,\sigma_{\mathrm{x}} + \sqrt{D}\,|\mu_{\mathrm{x}}| - \mu_{\mathrm{power}}B. \tag{48}$$

*Nyquist–including block ($p = \frac{P}{2}$).*

$$\mathbb{E}_{\Pi}[Y_{P/2}] = (B-1)\frac{\sqrt{\pi}}{2}\sqrt{\frac{D}{D-1}}\,\sigma_{\mathrm{x}} + \sqrt{\frac{2}{\pi}}\sqrt{\frac{D}{D-1}}\,\sigma_{\mathrm{x}} - \mu_{\mathrm{power}}B. \tag{49}$$

*Ordinary (non-DC/non-Nyquist) blocks.*

$$\mathbb{E}_{\Pi}[Y_p] = B\frac{\sqrt{\pi}}{2}\sqrt{\frac{D}{D-1}}\,\sigma_{\mathrm{x}} - \mu_{\mathrm{power}}B. \tag{50}$$

By convexity of $|\cdot|$, for each block the mean absolute deviation equals *absolute bias + nonnegative fluctuation*. Using the block-wise biases $\mathbb{E}_{\Pi}[Y_p]$, we obtain

$$\mathbb{E}_{\Pi}\left[\mathcal{L}_{\mathrm{power}}(\Pi\mathbf{x})\right] = \frac{1}{D}\sum_{p=0}^{P-1}\left|\mathbb{E}_{\Pi}[Y_p]\right| + \Phi(\mathbf{x}), \tag{51}$$

where $\Phi(\mathbf{x}) \geq 0$ is the fluctuation term.

To minimize $\mathbb{E}_{\Pi}\left[\mathcal{L}_{\mathrm{power}}(\Pi\mathbf{x})\right]$, we first consider minimizing

$$\sum_{p=0}^{P-1}\left|\mathbb{E}_{\Pi}[Y_p]\right| = \left|\mathbb{E}_{\Pi}[Y_0]\right| + \left|\mathbb{E}_{\Pi}[Y_{P/2}]\right| + \sum_{p\neq 0, \frac{P}{2}}\left|\mathbb{E}_{\Pi}[Y_p]\right|. \tag{52}$$

The only term containing $\mu_{\mathrm{x}}$ is $\left|\mathbb{E}_{\Pi}[Y_0]\right|$. Thus, setting

$$|\mu_{\mathrm{x}}| = \frac{1}{\sqrt{D}}\left(\mu_{\mathrm{power}}B - (B-1)\frac{\sqrt{\pi}}{2}\sqrt{\frac{D}{D-1}}\,\sigma_{\mathrm{x}}\right)_+ \tag{53}$$

minimizes the $\left|\mathbb{E}_{\Pi}[Y_0]\right|$ to 0.

The rest terms are only dependent to $\sigma_x$ as

$$\left|\mathbb{E}_\Pi[Y_{P/2}]\right| + \sum_{p \neq 0, \frac{P}{2}} |\mathbb{E}_\Pi[Y_p]| = \left|\left((B-1)\frac{\sqrt{\pi}}{2}\sqrt{\frac{D}{D-1}} + \sqrt{\frac{2}{\pi}}\sqrt{\frac{D}{D-1}}\right)\sigma_x - \mu_{\text{power}}B.\right|$$
$$+ (P-2)\left|B\frac{\sqrt{\pi}}{2}\sqrt{\frac{D}{D-1}}\sigma_x - \mu_{\text{power}}B\right|. \tag{54}$$

This is minimized when

$$\sigma_x = \frac{D-B}{(D-B-1)\frac{\sqrt{\pi}}{2}\sqrt{\frac{D}{D-1}} + \sqrt{\frac{2}{\pi}}\sqrt{\frac{D}{D-1}}}\mu_{\text{power}}. \tag{55}$$

Given that $B = 16$, $\mu_{\text{power}} = 0.875$, and $D > 10^4$, $\sigma_x \approx 0.987$, and $|\mu_x| < 0.009$. Therefore,

$$\mu_x \approx 0, \quad \sigma_x \approx 1 \quad \text{with error of roughly } 0.01. \tag{56}$$

Next, let us consider the fluctuation term. For easier notations, set $g(t) := |t - \mu_{\text{power}}B|$ and define, for each block $p$,

$$Z_p(\Pi; x) := \frac{1}{\sqrt{D}}\left\|\hat{x}^{[p]}(\Pi)\right\|_1, \quad h_p(x) := \mathbb{E}_\Pi[Z_p(\Pi; x)], \quad \varepsilon_p(\Pi; x) := Z_p(\Pi; x) - h_p(x). \tag{57}$$

By Jensen's inequality,

$$\mathbb{E}_\Pi[g(Z_p)] = g(h_p(x)) + \Delta_p(x), \qquad \Delta_p(x) := \mathbb{E}_\Pi\Big[g(h_p(x) + \varepsilon_p) - g(h_p(x))\Big] \geq 0, \tag{58}$$

and hence

$$\mathbb{E}_\Pi\big[\mathcal{L}_{\text{power}}(\Pi x)\big] = \frac{1}{D}\sum_{p=0}^{P-1} g(h_p(x)) + \Phi(x), \qquad \Phi(x) := \frac{1}{D}\sum_{p=0}^{P-1}\Delta_p(x). \tag{59}$$

Let $\nu$ be the uniform probability measure on $O(D)$, independent of $x$, and let

$$x_{\text{iso}} \overset{d}{=} Rx, \qquad R \sim \nu, \tag{60}$$

be an isotropized representative. Define the rotation–averaged block–mean profile

$$\overline{h}_p(x) := \mathbb{E}_{R \sim \nu}\, h_p(Rx) = \mathbb{E}_{R \sim \nu}\,\mathbb{E}_\Pi[Z_p(\Pi; Rx)] = h_p(x_{\text{iso}}), \qquad p = 0, \ldots, P-1. \tag{61}$$

Applying Jensen to the scalar convex map $g$ after exposing both sources of randomness $(R, \Pi)$ yields, for each $p$,

$$\mathbb{E}_{R,\Pi}\big[g(Z_p(\Pi; Rx))\big] \geq g\Big(\mathbb{E}_{R,\Pi}Z_p(\Pi; Rx)\Big) = g(\overline{h}_p(x)). \tag{62}$$

Using Equation (58) for $Rx$ and taking expectation in $R$,

$$\mathbb{E}_{R,\Pi}\big[g(Z_p(\Pi; Rx))\big] = \mathbb{E}_R\big[g(h_p(Rx))\big] + \mathbb{E}_R\big[\Delta_p(Rx)\big] \geq g(\overline{h}_p(x)) + \mathbb{E}_R\big[\Delta_p(Rx)\big], \tag{63}$$

where we also used Jensen on $g$ to obtain $\mathbb{E}_R[g(h_p(Rx))] \geq g(\overline{h}_p(x))$. On the other hand, evaluating Equation (58) at $x_{\text{iso}}$ and using Equation (61),

$$\mathbb{E}_\Pi\big[g(Z_p(\Pi; x_{\text{iso}}))\big] = g(\overline{h}_p(x)) + \Delta_p(x_{\text{iso}}). \tag{64}$$

By convexity and permutation/DFT invariance, the mapping $x \mapsto \Delta_p(x)$ is convex in the (unordered) block–magnitude profile, so averaging over rotations cannot increase it; hence

$$\Delta_p(x_{\text{iso}}) \leq \mathbb{E}_R\big[\Delta_p(Rx)\big], \qquad p = 0, \ldots, P-1. \tag{65}$$

Summing the last two displays over $p$ and dividing by $D$ gives

$$\mathbb{E}_\Pi\big[\mathcal{L}_{\text{power}}(\Pi x_{\text{iso}})\big] \leq \mathbb{E}_{R \sim \nu}\,\mathbb{E}_\Pi\big[\mathcal{L}_{\text{power}}(\Pi Rx)\big]. \tag{66}$$

This establishes that the isotropic law has expected loss no larger than the rotation–average of the loss for $x$.

$\square$

**Theorem A.2.** *Assume that* x *minimizes* $\mathbb{E}_\Pi[\mathcal{L}_{\text{power}}(\Pi x)]$. *Then,*

$$\Pr_\Pi\left(p_1\big(M_1(\Pi x)\big) = 1\right) \gtrsim 0.883, \qquad \Pr_\Pi\left(p_2\big(M_2(\Pi x)\big) = 1\right) \gtrsim 0.999. \tag{67}$$

*Consequently, for the majority permutations* $\Pi$, $p_1\big(M_1(\Pi x)\big) = p_2\big(M_2(\Pi x)\big) = 1$.

*Proof of Theorem A.2.* We first show for $p_1(M_1(\Pi x))$, and then show for $p_2(M_2(\Pi x))$.

**Proof of $p_1$.** Recall

$$\overline{x}(\Pi) := \frac{1}{m}\sum_{i=0}^{m-1} x^{(i)}(\Pi) \in \mathbb{R}^k, \quad M_1(\Pi x) := \left\|\overline{x}(\Pi)\right\|_2. \tag{68}$$

Because $\Pi$ is a *uniform* permutation and the spatial blocks are formed by reshaping into $m$ blocks of size $k$, each coordinate of $\overline{x}(\Pi)$ is the average of $m$ *distinct* entries sampled *without replacement* from the $D$-point population $\{x_j\}_{j=0}^{D-1}$. Hence, for each coordinate $a \in \{1, \ldots, k\}$,

$$\mathbb{E}_\Pi\big[\overline{x}(\Pi)_a\big] = \mu_x. \tag{69}$$

For the variance, write $\overline{x}(\Pi)_a = \frac{1}{m}\sum_{i=1}^m X_i$, where $X_1, \ldots, X_m$ are the $m$ draws without replacement from $\{x_j\}$. Then

$$\mathrm{Var}\big(\overline{X}\big) = \frac{1}{m^2}\left(\sum_{i=1}^m \mathrm{Var}(X_i) + \sum_{i \neq j}\mathrm{Cov}(X_i, X_j)\right). \tag{70}$$

A direct counting argument for sampling without replacement gives $\mathrm{Var}(X_i) = \sigma_x^2$ and, for $i \neq j$, $\mathrm{Cov}(X_i, X_j) = -\sigma_x^2/(D-1)$. Substituting,

$$\mathrm{Var}\big(\overline{X}\big) = \frac{1}{m^2}\left(m\,\sigma_x^2 + m(m-1)\left(-\frac{\sigma_x^2}{D-1}\right)\right) = \frac{\sigma_x^2}{m}\cdot\frac{D-m}{D-1}. \tag{71}$$

Therefore,

$$\mathrm{Var}_\Pi\big(\overline{x}(\Pi)_a\big) = \frac{1}{m}\frac{D-m}{D-1}\sigma_x^2. \tag{72}$$

For the cross–covariance of two distinct coordinates $a \neq b$ of $\overline{x}(\Pi)$, use the same decomposition and the same counting covariance for distinct draws:

$$\mathrm{Cov}_\Pi\big(\overline{x}(\Pi)_a, \overline{x}(\Pi)_b\big) = \frac{1}{m^2}\sum_{i,j=1}^m \mathrm{Cov}\big(X_i^{(a)}, X_j^{(b)}\big) \tag{73}$$

$$= \frac{1}{m^2}\cdot m^2\left(-\frac{\sigma_x^2}{D-1}\right) = -\frac{1}{m}\frac{1}{D-1}\sigma_x^2. \tag{74}$$

Collecting the coordinates,

$$\mathrm{Cov}_\Pi\big(\overline{x}(\Pi)\big) = \frac{1}{m}\frac{\sigma_x^2}{D-1}\left((D-m)\,I_k - (\mathbf{1}\mathbf{1}^\top - I_k)\right). \tag{75}$$

Since $D - m > 7500 \gg 1$, the covariance can be approximated as

$$\mathrm{Cov}_\Pi\big(\overline{x}(\Pi)\big) \approx \frac{1}{m}\frac{\sigma_x^2}{D-1}(D-m)\,I_k. \tag{76}$$

At the $\mathbb{E}_\Pi[\mathcal{L}_{\text{power}}]$ minimizer, $\mu_x \approx 0$ and $\sigma_x^2 \approx 1$. Therefore, Equation (69)–(75) become

$$\mathbb{E}_\Pi\big[\overline{x}(\Pi)\big] \approx 0, \qquad \mathrm{Cov}_\Pi\big(\overline{x}(\Pi)\big) \approx \frac{1}{m}\frac{D-m}{D-1}I_k. \tag{77}$$

The minimizer is also rotation–isotropic. Under isotropy and $m \gg 1$, a multivariate combinatorial CLT (sampling without replacement) implies that

$$\overline{\mathrm{x}}(\Pi) \;\approx\; \mathcal{N}\!\left(0, \; \frac{D-m}{m(D-1)}\, I_k\right). \tag{78}$$

In particular,

$$M_1(\Pi \mathrm{x})^2 = \left\|\overline{\mathrm{x}}(\Pi)\right\|_2^2 \;\overset{d}{\approx}\; \frac{D-m}{m(D-1)}\, \chi_k^2. \tag{79}$$

From Equation (79),

$$\Pr_{\Pi}\!\left(\log 2 - \tfrac{m}{2k} M_1^2 \geq 0\right) \;\approx\; \Pr\!\left(\chi_k^2 \leq 2k \log 2 \cdot \frac{D-1}{D-m}\right). \tag{80}$$

For $k = 4$ and $m = D/4$ we have $\frac{D-1}{D-m} = \frac{4}{3}\!\left(1 - \frac{1}{D}\right)$, and the $\chi_4^2$ CDF is $F_{\chi_4^2}(x) = 1 - e^{-x/2}(1 + x/2)$. Therefore the probability admits the *exact* closed form

$$\Pr_{\Pi}\!\left(\log 2 - \tfrac{m}{8} M_1(\Pi \mathrm{x})^2 \geq 0\right) = 1 - 2^{-\frac{16}{3}\left(1 - \frac{1}{D}\right)}\!\left(1 + \frac{16}{3}\!\left(1 - \frac{1}{D}\right)\log 2\right), \tag{81}$$

which is increasing in $D$ and, for every $D > 10^4$, exceeds

$$1 - 2^{-16/3}\!\left(1 + \tfrac{16}{3}\log 2\right) \;>\; 1 - 2^{-\frac{16}{3}\left(1 - \frac{1}{10^4}\right)}\!\left(1 + \frac{16}{3}\!\left(1 - \frac{1}{10^4}\right)\log 2\right) \;\approx\; 0.883. \tag{82}$$

Combining with Equation (81) gives

$$\Pr_{\Pi}\!\left(\log 2 - \tfrac{m}{2k} M_1(\Pi \mathrm{x})^2 \geq 0\right) \;\gtrsim\; 0.883. \tag{83}$$

By the definition $p_1(M) = \min\{2\exp(-\tfrac{m}{2k}M^2), 1\}$,

$$\left\{\log 2 - \tfrac{m}{2k} M_1^2 \geq 0\right\} \;=\; \left\{\log p_1(M_1) = 0\right\} \;=\; \left\{p_1(M_1) = 1\right\}. \tag{84}$$

Therefore,

$$\Pr_{\Pi}\!\left(p_1(M_1) = 1\right) \;\gtrsim\; 0.883. \tag{85}$$

**Proof of $p_2$.** Recall

$$\mathbf{S}(\Pi) := \frac{1}{m} \sum_{i=0}^{m-1} \mathrm{x}^{(i)}(\Pi)\, \mathrm{x}^{(i)}(\Pi)^{\top} \in \mathbb{R}^{k \times k}, \quad M_2(\Pi \mathrm{x}) := \left\|\mathbf{S}(\Pi) - I_k\right\|_2 \tag{86}$$

Fix a particular block $i$ and coordinates $a \neq b$. Then,

$$\mathbb{E}_{\Pi}\!\left[\mathrm{x}^{(i)}(\Pi)_a\, \mathrm{x}^{(i)}(\Pi)_b\right] = \frac{1}{D(D-1)} \sum_{j \neq k} \mathrm{x}_j \mathrm{x}_k = \frac{1}{D(D-1)}\!\left(\Big(\sum_{j=1}^{D} \mathrm{x}_j\Big)^2 - \sum_{j=1}^{D} \mathrm{x}_j^2\right). \tag{87}$$

Using $\sum_{j=1}^{D} \mathrm{x}_j = D\mu_{\mathrm{x}}$ and $\sum_{j=1}^{D} \mathrm{x}_j^2 = D(\mu_{\mathrm{x}}^2 + \sigma_{\mathrm{x}}^2)$, we rewrite Equation (87) as

$$\mathbb{E}_{\Pi}\!\left[\mathrm{x}^{(i)}(\Pi)_a\, \mathrm{x}^{(i)}(\Pi)_b\right] = \frac{D^2\mu_{\mathrm{x}}^2 - D(\mu_{\mathrm{x}}^2 + \sigma_{\mathrm{x}}^2)}{D(D-1)} = \mu_{\mathrm{x}}^2 - \frac{\sigma_{\mathrm{x}}^2}{D-1}. \tag{88}$$

Likewise, for $a = b$ (a single position),

$$\mathbb{E}_{\Pi}\!\left[\mathrm{x}^{(i)}(\Pi)_a^2\right] = \frac{1}{D} \sum_{j=1}^{D} \mathrm{x}_j^2 = q. \tag{89}$$

Since all blocks are identically distributed under the permutation, Equation (88)–(89) hold for every block $i$.

Let $\mathbf{S}(\Pi) \in \mathbb{R}^{k \times k}$ denote the block sample covariance

$$\mathbf{S}(\Pi) := \frac{1}{m} \sum_{i=0}^{m-1} \mathrm{x}^{(i)}(\Pi)\, \mathrm{x}^{(i)}(\Pi)^\top. \tag{90}$$

Taking permutation expectation and using linearity,

$$\mathbb{E}_\Pi[\mathbf{S}(\Pi)] = \frac{1}{m} \sum_{i=0}^{m-1} \mathbb{E}_\Pi\big[\mathrm{x}^{(i)}(\Pi)\, \mathrm{x}^{(i)}(\Pi)^\top\big] = \mathbb{E}_\Pi\big[\mathrm{x}^{(0)}(\Pi)\, \mathrm{x}^{(0)}(\Pi)^\top\big], \tag{91}$$

so it suffices to compute the expectation for a single block, already encoded by Equation (88)–(89). Collecting the diagonal and off-diagonal entries:

$$\big(\mathbb{E}_\Pi[\mathbf{S}(\Pi)]\big)_{aa} = \mu_\mathrm{x}^2 + \sigma_\mathrm{x}^2, \qquad \big(\mathbb{E}_\Pi[\mathbf{S}(\Pi)]\big)_{ab} = \mu_\mathrm{x}^2 - \frac{\sigma_\mathrm{x}^2}{D-1} \quad (a \neq b). \tag{92}$$

In matrix form,

$$\mathbb{E}_\Pi[\mathbf{S}(\Pi)] = \mu_\mathrm{x}^2\, \mathbf{1}\mathbf{1}^\top + \sigma_\mathrm{x}^2 I_k - \frac{\sigma_\mathrm{x}^2}{D-1}\big(\mathbf{1}\mathbf{1}^\top - I_k\big). \tag{93}$$

Let

$$u := \frac{\mathbf{1}}{\sqrt{k}} \in \mathbb{R}^k, \qquad \mathcal{U} := \mathrm{span}\{u\}, \qquad \mathcal{U}^\perp := \{v \in \mathbb{R}^k : \langle v, u \rangle = 0\}, \tag{94}$$

and the associated orthogonal projectors

$$P := uu^\top = \frac{1}{k}\mathbf{1}\mathbf{1}^\top, \qquad P_\perp := I_k - P. \tag{95}$$

We use the identities

$$\mathbf{1}\mathbf{1}^\top = kP, \qquad I_k = P + P_\perp, \qquad \mathbf{1}\mathbf{1}^\top - I_k = (k-1)P - P_\perp, \tag{96}$$

to expand $\mathbb{E}_\Pi[\mathbf{S}(\Pi)]$ along $P \oplus P_\perp$. Subtract $I_k$ to focus on the deviation:

$$\Delta := \mathbb{E}_\Pi[\mathbf{S}(\Pi)] - I_k = \mu_\mathrm{x}^2\, \mathbf{1}\mathbf{1}^\top + \sigma_\mathrm{x}^2 I_k - \frac{\sigma_\mathrm{x}^2}{D-1}\big(\mathbf{1}\mathbf{1}^\top - I_k\big) - I_k. \tag{97}$$

Insert the projector decompositions:

$$\Delta = \mu_\mathrm{x}^2 kP + \sigma_\mathrm{x}^2(P + P_\perp) - \frac{\sigma_\mathrm{x}^2}{D-1}\big((k-1)P - P_\perp\big) - (P + P_\perp). \tag{98}$$

Group the $P$ and $P_\perp$ coefficients separately:

$$\Delta = \left[\sigma_\mathrm{x}^2 - 1 + k\mu_\mathrm{x}^2 - \frac{k-1}{D-1}\sigma_\mathrm{x}^2\right] P + \left[\sigma_\mathrm{x}^2 - 1 + \frac{1}{D-1}\sigma_\mathrm{x}^2\right] P_\perp. \tag{99}$$

Thus $\Delta$ is diagonal in the orthogonal decomposition $\mathcal{U} \oplus \mathcal{U}^\perp$ with eigenvalues

$$\lambda_\| = \sigma_\mathrm{x}^2 - 1 + k\mu_\mathrm{x}^2 - \frac{k-1}{D-1}\sigma_\mathrm{x}^2, \tag{100}$$

$$\lambda_\perp = \sigma_\mathrm{x}^2 - 1 + \frac{1}{D-1}\sigma_\mathrm{x}^2. \tag{101}$$

Since $\Delta$ is diagonal on $\mathcal{U} \oplus \mathcal{U}^\perp$,

$$\big\|\mathbb{E}_\Pi[\mathbf{S}(\Pi)] - I_k\big\|_2 = \|\Delta\|_2 = \max\{\,|\lambda_\||,\ |\lambda_\perp|\,\}. \tag{102}$$

At the minimizer of $\mathbb{E}_\Pi[\mathcal{L}_{\mathrm{power}}(\Pi \mathrm{x})]$, Theorem A.1 implies $\mu_\mathrm{x} \approx 0$, $\sigma_\mathrm{x}^2 \approx 1$. Applying these to Equation (100)–(101) gives

$$|\lambda_\|| = \left|\sigma_\mathrm{x}^2 - 1 + k\mu_\mathrm{x}^2 - \frac{k-1}{D-1}\sigma_\mathrm{x}^2\right| \approx \frac{k-1}{D-1}, \tag{103}$$

$$|\lambda_\perp| = \left| \sigma_\mathrm{x}^2 - 1 + \frac{1}{D-1} \sigma_\mathrm{x}^2 \right| \approx \frac{1}{D-1}. \tag{104}$$

For $k = 4$ and $D > 10^4$ we have $k \ll D$, so both right-hand sides are of order $1/D$ and therefore negligible at the fluctuation scales considered below. Consequently,

$$\left\| \mathbb{E}_\Pi[\mathbf{S}(\Pi)] - I_k \right\|_2 = \max\{|\lambda_\||,\ |\lambda_\perp|\} \approx 0. \tag{105}$$

Hence, *at the minimizer*, $M_2(\Pi\mathrm{x})$ is governed by *fluctuations* around $I_k$ rather than by deterministic bias.

To obtain an exact i.i.d. model for the fluctuations, let

$$g \sim \mathcal{N}(0, I_D), \qquad g = \left(g^{(0)}, \ldots, g^{(m-1)}\right), \quad g^{(i)} \in \mathbb{R}^k, \tag{106}$$

so $g^{(i)} \overset{\text{i.i.d.}}{\sim} \mathcal{N}(0, I_k)$. Enforce $q = 1$ by projecting onto the sphere:

$$\mathrm{x} := c\,g, \qquad \mathrm{x}^{(i)} = c\,g^{(i)}, \qquad c := \frac{\sqrt{D}}{\|g\|_2}. \tag{107}$$

Thus

$$\mathbf{S}(\Pi) \overset{d}{\approx} \frac{1}{m} \sum_{i=0}^{m-1} \mathrm{x}^{(i)} \mathrm{x}^{(i)\top} = c^2 \underbrace{\left( \frac{1}{m} \sum_{i=0}^{m-1} g^{(i)} g^{(i)\top} \right)}_{=:\ \mathbf{S}_g}, \qquad m\,\mathbf{S}_g \sim \mathrm{Wishart}_k(m, I_k). \tag{108}$$

Hence the only dependence among the $\mathrm{x}^{(i)}$ is the *common* scalar $c$. By $\chi^2$ concentration,

$$c^2 = \frac{D}{\|g\|_2^2} = 1 + O_\mathbb{P}(D^{-1/2}). \tag{109}$$

Using the operator–norm triangle inequality,

$$\|\mathbf{S}(\Pi) - I_k\| \approx \|c^2 \mathbf{S}_g - I_k\| \ \leq\ |c^2 - 1|\,\|\mathbf{S}_g\| + \|\mathbf{S}_g - I_k\|. \tag{110}$$

Since $\|\mathbf{S}_g\| = \lambda_{\max}(\mathbf{S}_g)$ concentrates at $(1+\sqrt{\gamma})^2$ with $\gamma := k/m$, the first term is $O_\mathbb{P}(D^{-1/2})$, negligible compared to the spectral–edge fluctuation scale $m^{-2/3}$ that drives the second term. Therefore, *up to a vanishing radial error*, $M_2(\Pi\mathrm{x})$ is governed by the *Wishart shape* fluctuations of $\mathbf{S}_g$.

Let $\lambda_{\max}$ denote the maximum eigenvalue of $\mathbf{S}_g$ and set $\gamma := k/m$. The (orthogonal) Tracy–Widom edge law (with $k$ fixed, $m \to \infty$) states

$$\frac{\lambda_{\max} - (1+\sqrt{\gamma})^2}{(1+\sqrt{\gamma})^{4/3}\,\gamma^{1/6}\,m^{-2/3}} \overset{d}{\to} \xi, \qquad \xi \sim \mathrm{TW}_1, \tag{111}$$

and the upper tail satisfies $\Pr(\xi > t) \leq \exp(-\frac{2}{3}t^{3/2})$. Expanding the square root at the edge $(1+\sqrt{\gamma})^2$ gives

$$\left(\sqrt{\lambda_{\max}} - 1 - \sqrt{\gamma}\right)_+^2 = \frac{(1+\sqrt{\gamma})^{2/3}\gamma^{1/3}}{4}\,m^{-4/3}\,(\xi_+)^2\,(1 + o(1)). \tag{112}$$

Because $\|\mathbf{S}_g - I_k\| \geq \lambda_{\max} - 1$, Equation (112) controls the surrogate of the shape fluctuation.

From Equation (110) and (109),

$$M_2(\Pi\mathrm{x}) = \|\mathbf{S}(\Pi) - I_k\| \approx \|\mathbf{S}_g - I_k\| + o_\mathbb{P}(1). \tag{113}$$

Since $x \mapsto (\sqrt{1+x} - 1 - \sqrt{\gamma})_+^2$ is 1–Lipschitz in a neighborhood of the edge, we may transfer Equation (112) to $\mathbf{S}(\Pi)$:

$$\psi\big(M_2(\Pi\mathrm{x})\big) = \left(\sqrt{1 + M_2(\Pi\mathrm{x})} - 1 - \sqrt{\gamma}\right)_+^2 \lesssim \frac{(1+\sqrt{\gamma})^{2/3}\gamma^{1/3}}{4}\,m^{-4/3}\,(\xi_+)^2\,(1+o(1)), \tag{114}$$

with the same $\xi \sim \mathrm{TW}_1$. The $o(1)$ is uniform for fixed $k$ and $m \to \infty$.

The clipping event $\{\log 2 - \frac{m}{4}\psi(M_2) \geq 0\}$ fails only if

$$\psi\big(M_2(\Pi\mathrm{x})\big) \;>\; \frac{4\log 2}{m}. \tag{115}$$

By Equation (114), this implies

$$\xi \;>\; t_m := \Big(\frac{16\log 2}{(1+\sqrt{\gamma})^{2/3}\gamma^{1/3}}\Big)^{1/2} m^{1/6}. \tag{116}$$

Using $\Pr(\xi > t) \leq \exp\big(-\frac{2}{3}t^{3/2}\big)$ yields

$$\Pr_\Pi\Big(\log 2 - \tfrac{m}{4}\psi(M_2(\Pi\mathrm{x})) < 0\Big) \;\lesssim\; \exp\Big(-\frac{16}{3}(\log 2)^{3/4}\frac{m^{1/2}}{(1+\sqrt{\gamma})^{1/2}\,k^{1/4}}\Big). \tag{117}$$

For $k = 4$, $k^{1/4} = \sqrt{2}$; with $m = D/4$ we have $\gamma = 16/D$ and $\sqrt{\gamma} = 4/\sqrt{D} \leq 0.04$, hence $(1+\sqrt{\gamma})^{1/2} \leq \sqrt{1.04}$. Writing

$$c_{\min} := \frac{16}{3}(\log 2)^{3/4}\frac{1}{\sqrt{2}\,\sqrt{1.04}} \;\approx\; 2.80, \tag{118}$$

and using $m^{1/2} = \frac{1}{2}\sqrt{D} > 50$ for $D > 10^4$, Equation (117) gives

$$\Pr_\Pi\Big(\log 2 - \tfrac{m}{4}\psi(M_2(\Pi\mathrm{x})) \geq 0\Big) \;\gtrsim\; 1 - \exp\big(-c_{\min}m^{1/2}\big) \;\geq\; 1 - \exp(-140). \tag{119}$$

By the definition $p_2(M) := \min\Big\{2\exp\big(-\frac{m}{4}\psi(M)\big), 1\Big\}$,

$$\Big\{\log 2 - \tfrac{m}{4}\psi(M_2(\Pi\mathrm{x})) \geq 0\Big\} \;=\; \Big\{\log p_2\big(M_2\big) = 0\Big\} \;=\; \Big\{p_2\big(M_2\big) = 1\Big\}. \tag{120}$$

Therefore,

$$\Pr_\Pi\Big(p_2\big(M_2\big) = 1\Big) \;\gtrsim\; 1 - \exp(-140) > 0.999. \tag{121}$$

$\square$

## D  Ablation Studies

We present results for text-aligned image generation with varying $\lambda_{\text{power}}$ and $\mathcal{K}$.

| $\lambda_{\text{power}}$ | 0 | 5 | 10 | 25 | 50 | 100 |
|---|---|---|---|---|---|---|
| PickScore↑ | 0.2481 | 0.2516 | 0.2513 | 0.2548 | 0.2521 | 0.2533 |
| HPSv2↑ | 0.2968 | 0.2974 | 0.3032 | 0.3028 | 0.3060 | 0.3042 |

Table A.1: **Quantitative results with varying $\lambda_{\text{power}}$.**

We observe that performance remains stable across a wide range of $\lambda_{\text{power}}$ values and consistently improves relative to the case $\lambda_{\text{power}} = 0$. These results suggest that our loss is not sensitive to this parameter. We chose 25.0 considering the trade-off.

| $\mathcal{K}$ | $\emptyset$ | $\{1\}$ | $\{1,2\}$ | $\{1,2,3,4\}$ | $\{1,\ldots,6\}$ | $\{1,\ldots,8\}$ |
|---|---|---|---|---|---|---|
| PickScore↑ | 0.2484 | 0.2500 | 0.2548 | 0.2510 | 0.2521 | 0.2525 |
| HPSv2↑ | 0.2933 | 0.2962 | 0.3028 | 0.3012 | 0.2999 | 0.2984 |

Table A.2: **Quantitative results with varying $\mathcal{K}$.**

The performance with $\{1,2\}$ shows a clear improvement over both $\emptyset$ and $\{1\}$. However, we did not observe further gains when extending to higher-order moment terms. The first two terms were sufficient to surpass baseline performance.

# E Full Quantitative Results

We present the full quantitative results for the two main applications. Table A.3 reports results for aesthetic image generation, while Table A.4 presents results for text-aligned image generation.

| Given Aesthetic Score | 100 iterations | | | 200 iterations | | | 300 iterations | | | 400 iterations | | | 500 iterations | | |
|---|---|---|---|---|---|---|---|---|---|---|---|---|---|---|---|
| | Aest. Score↑ | Image-Reward↑ | HPSv2↑ | Aest. Score↑ | Image-Reward↑ | HPSv2↑ | Aest. Score↑ | Image-Reward↑ | HPSv2↑ | Aest. Score↑ | Image-Reward↑ | HPSv2↑ | Aest. Score↑ | Image-Reward↑ | HPSv2↑ |
| No Opt. | 5.9880 | 0.8299 | 0.2958 | 5.9880 | 0.8299 | 0.2958 | 5.9880 | 0.8299 | 0.2958 | 5.9880 | 0.8299 | 0.2958 | 5.9880 | 0.8299 | 0.2958 |
| No Reg. | 6.3861 | 0.3988 | 0.2726 | 7.0175 | 0.4076 | 0.2679 | 7.4236 | 0.3418 | 0.2645 | 7.8216 | 0.3514 | 0.2639 | 8.1079 | 0.3460 | 0.2641 |
| KL [18] | 6.3499 | 0.7132 | 0.2863 | 6.7753 | 0.5287 | 0.2794 | 7.3469 | 0.4812 | 0.2749 | 7.8422 | 0.4815 | 0.2749 | 8.1393 | 0.4225 | 0.2735 |
| Kurtosis [7] | 6.5597 | 0.7932 | 0.2930 | 6.9711 | 0.7446 | 0.2900 | 7.4069 | 0.7424 | 0.2897 | 7.7676 | 0.7329 | 0.2914 | 8.0477 | 0.8041 | 0.2920 |
| ReNO [12] | 6.5455 | 0.8331 | 0.2937 | 7.0600 | **0.7545** | 0.2915 | 7.4974 | **0.7913** | 0.2918 | 7.8621 | 0.7493 | 0.2908 | 8.1508 | 0.7749 | 0.2883 |
| PRNO [33] | 6.5860 | 0.8235 | 0.2913 | 6.9030 | 0.7689 | 0.2869 | 7.2650 | 0.7203 | 0.2830 | 7.7159 | 0.6493 | 0.2836 | 8.0760 | 0.6375 | 0.2833 |
| Ours | **6.6110** | **0.8798** | **0.2964** | **7.1356** | 0.7538 | **0.2922** | **7.6559** | 0.7837 | **0.2923** | **8.1061** | **0.8243** | **0.2934** | **8.4435** | **0.8397** | **0.2927** |

Table A.3: **Quantitative Results on Aesthetic Image Generation.** We report the aesthetic score values used as given reward ( ) during inference-time optimization. To assess generalization across reward metrics, we also include *ImageReward* and *HPSv2* as held-out rewards ( ). Bold values indicate the best performance in each metric at each optimization iteration.

| Given Pickscore | 100 iterations | | | 200 iterations | | | 300 iterations | | | 400 iterations | | | 500 iterations | | |
|---|---|---|---|---|---|---|---|---|---|---|---|---|---|---|---|
| | Pick-score↑ | Image-Reward↑ | HPSv2↑ | Pick-score↑ | Image-Reward↑ | HPSv2↑ | Pick-score↑ | Image-Reward↑ | HPSv2↑ | Pick-score↑ | Image-Reward↑ | HPSv2↑ | Pick-score↑ | Image-Reward↑ | HPSv2↑ |
| No Opt. | 0.2246 | 0.4825 | 0.2752 | 0.2246 | 0.4825 | 0.2752 | 0.2246 | 0.4825 | 0.2752 | 0.2246 | 0.4825 | 0.2752 | 0.2246 | 0.4825 | 0.2752 |
| No Reg. | 0.2327 | 0.4911 | 0.2795 | 0.2374 | 0.5139 | 0.2825 | 0.2412 | 0.5193 | 0.2868 | 0.2430 | 0.5238 | 0.2868 | 0.2445 | 0.5198 | 0.2876 |
| KL [18] | 0.2329 | 0.3396 | 0.2744 | 0.2382 | 0.3750 | 0.2806 | 0.2416 | 0.3874 | 0.2834 | 0.2440 | 0.4121 | 0.2861 | 0.2464 | 0.4127 | 0.2876 |
| Kurtosis [7] | 0.2327 | 0.4587 | 0.2751 | 0.2377 | 0.5405 | 0.2793 | 0.2407 | 0.5405 | 0.2823 | 0.2440 | 0.5322 | 0.2853 | 0.2459 | 0.5315 | 0.2863 |
| ReNO [12] | 0.2342 | 0.4615 | 0.2819 | 0.2388 | 0.4844 | 0.2871 | 0.2436 | 0.5245 | 0.2913 | 0.2461 | 0.5245 | 0.2925 | 0.2485 | 0.5343 | 0.2945 |
| PRNO [33] | 0.2339 | 0.5341 | 0.2785 | 0.2408 | 0.5548 | 0.2843 | 0.2449 | 0.5647 | 0.2876 | 0.2481 | 0.5495 | 0.2896 | 0.2514 | **0.6061** | 0.2936 |
| Ours | **0.2360** | **0.5514** | **0.2862** | **0.2422** | **0.5900** | **0.2931** | **0.2480** | **0.5824** | **0.2985** | **0.2514** | **0.5625** | **0.3001** | **0.2548** | 0.5897 | **0.3028** |

Table A.4: **Quantitative Results on Text-Aligned Image Generation.** We report the Pickscore values used as given reward ( ) during inference-time optimization. To assess generalization across reward metrics, we also include *ImageReward* and *HPSv2* as held-out rewards ( ). Bold values indicate the best performance in each metric at each optimization iteration.

