# OpenReview forum: "Moment- and Power-Spectrum-Based Gaussianity Regularization for Text-to-Image Models"
_NeurIPS.cc/2025/Conference — NeurIPS 2025 poster_

### Official Review · Reviewer_mYPt · 2025-07-01

**Clarity:** 3
**Significance:** 2
**Originality:** 3
**Rating:** 4
**Confidence:** 3

**Summary:**

1, This paper proposes a unified regularization framework for latent optimization in text-to-image generative models. Authors combine moment-based regularization in the spatial domain with power-spectrum-based regularization in the frequency domain to better enforce standard Gaussianity of latent variables.

2. This approach is motivated by the limitations of existing regularizers (e.g., KL, norm, or kurtosis penalties), which may fail to eliminate residual structures or correlations in the latent space. By encouraging the latent to match both the marginal moment statistics and the expected power spectrum of white Gaussian noise, the method prevents reward hacking and improves optimization stability during test-time reward alignment.

3, Experiments on aesthetic and text-alignment tasks with a one-step text-to-image model show that the proposed method achieves higher reward objectives while maintaining better image quality compared to existing approaches.

**Questions:**

Would the method generalize to other text-to-image model except FLUX, multi-step diffusion models or text-to-video models? It would be good to see the discussion.

Is there more quantitative / qualitative results that support the conclusion? Right now, the experiments are very toy.

**Ethical Concerns:**

["NO or VERY MINOR ethics concerns only"]

**Final Justification:**

The author has addressed my concerns.

**Limitations:**

1.it would be good to expand on how latent regularization might propagate biases from the pre-trained generator.

2. Only on a single text-to-image model (FLUX) with limited prompts are mainly evaluated.

**Paper Formatting Concerns:**

No.

**Quality:**

3

**Strengths And Weaknesses:**

Strength:

1. The method is well-motivated and theoretically grounded, and very good written.

2. Reward hacking is well known issue and important topic.

Weakness:

1. This paper is mainly evaluated only on a single text-to-image model (FLUX) with limited prompts. Broader evaluation on other architectures or diffusion models would strengthen the paper.

2. It would help if authors can discuss more regard whether matching higher-order moments could yield further improvements or not.

3. It assumes that optimizing the latent alone suffices to align rewards, but i didnt see it explore interactions with jointly optimizing text or conditioning variables, which might be relevant in practice.

---

> ### Author Rebuttal · Authors · 2025-07-31
>
> We sincerely appreciate your time and thoughtful feedback. We are especially grateful for your recognition that our method is well-motivated, theoretically grounded, and addresses the critical challenge of reward hacking. We also value your observation regarding the complementary roles of spatial and spectral domain regularization in enforcing Gaussianity.
>
> We would like to take this opportunity to re-emphasize our contributions:
>
> - **Moment-based loss (Section 3.1):** Unifies independently developed regularization terms such as **KL divergence**, **norm**, and **kurtosis** losses within a single framework.
>
> - **Power-spectrum-based loss (Section 3.2):** Defined via the **DFT** in the spectral domain, it regularizes DFT coefficients to match their theoretical distribution. Unlike **PRNO**, which removes spatial correlation by aligning the empirical covariance matrix with the identity and incurs quadratic complexity, our method uses **FFT** for efficient computation with $O(D \log D)$ complexity and operates via a fundamentally different mechanism.
>
> - **Dual-domain regularization (Section 3.3):** We apply spatial and spectral losses **jointly**, leveraging their **complementary strengths**—the spatial loss ensures marginal distribution alignment, while the spectral loss enforces spatial neutrality. (**L116**)
>
> Below, we address the specific questions and concerns raised:
>
> ---
> ***Q:*** *This paper is mainly evaluated only on a single text-to-image model (FLUX) with limited prompts. Broader evaluation on other architectures or diffusion models would strengthen the paper*
>
> ***A:*** To assess the generality of our approach, we additionally evaluated our method using the Stable Diffusion XL-Turbo (SDXL-Turbo) model for text-aligned image generation:
>
> | SDXL Turbo | ReNO | PRNO | Ours |
> |---|---|---|---|
> | **PickScore**$\uparrow$ | 0.2564 | 0.2595 | **0.2628** |
> | **ImageReward**$\uparrow$ | 0.6735 | 0.6430 | **0.7063** |
>
> Our method **outperforms** the two strongest baselines on PickScore and ImageReward, demonstrating its effectiveness beyond the original FLUX model. We will include these results in the revised version of the paper.
>
> ---
> ***Q:*** *It would help if authors can discuss more regard whether matching higher-order moments could yield further improvements or not.*
>
> ***A:*** We report additional results using different $\mathcal{K}$ values in text-aligned image generation:
>
> | $\mathcal{K}$ | $\emptyset$| \{1\} | \{1,2\} | \{1,2,3,4\} | \{1,...,6\} | \{1,...,8\} |
> |---|---|---|---|---|---|---|
> | **PickScore**$\uparrow$ | 0.2484 | 0.2500 | 0.2548 | 0.2510 | 0.2521 | 0.2525 |
> | **HPSv2**$\uparrow$ | 0.2933 | 0.2962 | 0.3028 | 0.3012 | 0.2999 | 0.2984 |
>
> The performance with \{1,2\} shows a clear performance improvement over both $\emptyset$ and \{1\}. However, we did not observe further improvements when extending to higher-order moment losses. The first two terms were sufficient to surpass baseline performance, as our regularization loss was designed not only to incorporate moment-based losses but also to include a power-spectrum-based loss that effectively suppresses residual spatial structure. (**L187**) Additionally, we provide a unified framework that integrates regularization losses which had previously been used independently.
>
> ---
> ***Q:*** *It assumes that optimizing the latent alone suffices to align rewards, but I didn't see it explore interactions with jointly optimizing text or conditioning variables, which might be relevant in practice.*
>
> ***A:*** While we could not find prior work exploring joint optimization of conditional variables in test-time reward alignment for text-to-image models, we investigated this direction by updating both the text embedding and the latent variable during optimization. The results are summarized below:
>
> | No Reg. | No Emb. Opt. | Emb. Opt.| // | Ours | No Emb. Opt. | Emb. Opt.|
> |---|---|---|---|---|---|---|
> | **PickScore**$\uparrow$ | 0.2445 | 0.2436 | // | **PickScore**$\uparrow$ | 0.2548 | 0.2526 |
> | **HPSv2**$\uparrow$ | 0.2876 | 0.2802 | // | **HPSv2**$\uparrow$ | 0.3028 | 0.2912 |
>
> **Reg.** denotes regularization, and **Emb. Opt.** denotes text embedding optimization.
>
> We did not observe any performance improvement from text embedding joint optimization.
>
> ---
> Additionally, we would greatly appreciate it if you could kindly elaborate the following point you raised as a limitation:
>
> “*it would be good to expand on how latent regularization might propagate biases from the pre-trained generator.*”
>
> We are eager to understand your concern more deeply so that we may address it properly in the revised version.
>
> ---

---

> > ### Comment · Reviewer_mYPt · 2025-08-04
> >
> > Thank the author for the reply. It has addressed my concerns. I will raise my final score.

---

> ### Author Response · Authors · 2025-08-07
>
> We sincerely thank you for acknowledging our rebuttal and for indicating that our clarifications addressed the concerns, and we truly appreciate the decision to raise the final score.

---

### Official Review · Reviewer_p3aM · 2025-07-02

**Clarity:** 3
**Significance:** 3
**Originality:** 3
**Rating:** 5
**Confidence:** 3

**Summary:**

This paper proposes a novel regularization loss function to enforce standard Gaussianity in the latent space of text-to-image generative models. The method combines moment-based regularization in the spatial domain with power spectrum-based regularization in the spectral domain, treating high-dimensional samples as collections of i.i.d. standard Gaussian variables. The authors demonstrate that several existing Gaussianity regularization methods (KL divergence, kurtosis, norm-based regularization) can be viewed as special cases of their unified framework. In the spectral domain, they leverage the fact that the power spectrum of i.i.d. Gaussian samples follows a chi-square distribution to design corresponding regularization terms. The method shows superior performance in test-time reward alignment tasks for text-to-image models, effectively preventing reward hacking and accelerating convergence.

**Questions:**

+ How were key parameters like λpower=25.0 chosen? Can you provide parameter sensitivity analysis or adaptive parameter selection methods?

+ Given the acknowledged limitation that loss function values cannot reliably indicate Gaussianity, could you design better Gaussianity evaluation metrics? This is important for practical applications.

+ How does the method perform on other generative models (e.g., Stable Diffusion, DALL-E)?

**Ethical Concerns:**

["NO or VERY MINOR ethics concerns only"]

**Final Justification:**

After reviewing the rebuttal and discussions, I am inclined to support this paper. Given its originality and relevance, I believe this paper could be a strong candidate for acceptance at NeurIPS.

**Limitations:**

Yes

**Paper Formatting Concerns:**

None.

**Quality:**

3

**Strengths And Weaknesses:**

Strengths:

+ The paper provides a unified theoretical framework that encompasses multiple existing Gaussianity regularization methods under the moment-matching paradigm, with Theorem 1 providing a rigorous characterization of standard Gaussian distributions via moment conditions.

+ The combination of spatial and spectral domain regularization is innovative, particularly the use of DFT coefficient distributions to design spectral domain regularization.

+ Compared to PRNO's O(D²) complexity, the method achieves O(D log D) time complexity through FFT, significantly improving efficiency.

+ The method demonstrates superior performance on both aesthetic image generation and text-aligned image generation tasks, with detailed ablation studies.

Weaknesses:

+ While the paper provides theoretical foundations for moment conditions and spectral properties, it lacks deeper theoretical analysis of why dual-domain regularization is necessary.

+ Key parameters like λpower=25.0 lack theoretical justification or systematic hyperparameter analysis.

+ The authors acknowledge that their loss function value alone cannot reliably indicate how closely a latent vector matches a true standard Gaussian distribution, which is a significant limitation.

+ Experiments are primarily conducted on the FLUX model, with insufficient validation of generalization to other generative models.

---

> ### Author Rebuttal · Authors · 2025-07-31
>
> Thank you for taking the time to review our work and provide thoughtful and constructive feedback. We organize our responses into four categories:
>
> ---
> ***Q:*** *While the paper provides theoretical foundations for moment conditions and spectral properties, it lacks deeper theoretical analysis of why dual-domain regularization is necessary.*
>
> ***A:*** Our core rationale for dual-domain regularization is rooted in the complementary limitations and strengths of moment-based and power-spectrum-based losses—an insight supported theoretically.
>
> Moment-based losses are **permutation-invariant** and therefore do not account for the **ordering** of elements in a sequence. (**L185**) For instance, consider a sorted sequence of 10K samples drawn from a standard Gaussian distribution. Although this sequence does not resemble a natural realization of random Gaussian samples—since such perfect ordering is highly improbable—the moment-based loss cannot distinguish it from a genuinely random Gaussian sample because it only matches marginal statistics. This limitation is theoretical, not merely empirical.
>
> To address this, we propose a power-spectrum-based loss, which is **permutation-variant** and sensitive to spatial correlations. (**L43**, **L186**) By analyzing the DFT coefficients, it captures deviations from white Gaussian noise and penalizes residual spatial structure. In **Lemma 1**, we provide a theoretical characterization of the expected distribution of the power spectrum under the assumption of independently sampled Gaussian noise in the spatial domain. This formal foundation justifies the use of spectral loss as a principled way to enforce spatial neutrality.
>
> However, the **spectral loss alone** does not ensure alignment with the target **marginal distribution**, as it focuses only on spatial structure. Therefore, **both losses must be used together**:
>
> - The **moment-based loss** ensures the correct marginal distribution.
>
> - The **spectral loss** enforces the absence of spatial correlation.
>
> These two losses play **complementary roles** in fully regularizing toward Gaussianity (**L116**). To further support this, we provide visual evidence in **Figure 1**, which illustrates why regularization in both domains is necessary.
>
> If you have a specific theoretical direction or additional depth in mind, we would greatly appreciate further guidance and are open to incorporating it into the revision.
>
>
> ---
> ***Q:*** *How were key parameters like λpower=25.0 chosen?*
>
> ***A:*** We present results on text-aligned image generation using different values of $\lambda_{\text{power}}$:
>
> | $\lambda_{\text{power}}$| 0 | 5 | 10 | 25 | 50 | 100 |
> |---|---|---|---|---|---|---|
> | **PickScore**$\uparrow$ | 0.2481 | 0.2516 | 0.2513 | 0.2548 | 0.2521 | 0.2533 |
> | **HPSv2**$\uparrow$ | 0.2968 | 0.2974 | 0.3032 | 0.3028 | 0.3060 | 0.3042 |
>
> We observe that performance remains stable across a wide range of $\lambda_{\text{power}}$ values and consistently improves over the case with $\lambda_{\text{power}} = 0$. These results suggest that our loss is not sensitive to this parameter. We chose 25.0 considering the trade-off.
>
> ---
> ***Q:*** *Loss function values cannot reliably indicate Gaussianity, could you design better Gaussianity evaluation metrics?*
>
> ***A:*** We would like to clarify the intent of our statement in L327. The key point of L327 is that while our loss is effective for regularization—optimizing it guides the latent toward the Gaussian manifold (**Figure 3**) and prevents deviation—its absolute value is not meant to serve as a global Gaussianity score for comparing arbitrarily distant points. As the title "**Gaussianity Regularization**" suggests, our focus is on **designing an effective regularization**, not a general-purpose metric. This distinction does not diminish its utility, as our experimental results consistently support its effectiveness in practice.
>
> ---
> ***Q:*** *How does the method perform on other generative models?*
>
> ***A:*** We evaluated our method using the Stable Diffusion XL-Turbo (SDXL-Turbo) model for text-aligned image generation:
>
> | SDXL Turbo | ReNO | PRNO | Ours |
> |---|---|---|---|
> | **PickScore**$\uparrow$ | 0.2564 | 0.2595 | **0.2628** |
> | **ImageReward**$\uparrow$ | 0.6735 | 0.6430 | **0.7063** |
>
> Our method achieves **improved performance** over the two strongest baselines in both PickScore and ImageReward metrics, demonstrating its effectiveness beyond the originally tested model, FLUX. We will include this result in the revised version of the paper.
>
> ---

---

> > ### Author Response · Authors · 2025-08-07
> >
> > We appreciate your time throughout the review process. As the discussion period is drawing to a close, please let us know if there are any final questions or points you would like us to address. We remain available for any remaining feedback.

---

### Official Review · Reviewer_LvLi · 2025-07-05

**Clarity:** 1
**Significance:** 3
**Originality:** 3
**Rating:** 5
**Confidence:** 3

**Summary:**

- Authors propose a spatial and spectral regularization loss that enforces gaussianity in latents used for reward alignment in text to image models.
- In the spatial domain, authors treat a N dimensional gaussian as a collection of 1-D iid gaussian values and utilize the moments of the samples to define a moment-based regularization loss. Authors show that the commonly used KL-divergence, Kurtosis and Norm regularizations are special cases of this spatial domain gaussianity regularization loss.
- Authors observed that while the latent can have correct marginals in the spatial domain but there can be a mismatch in the spectral domain. To correct this, authors propose another regularization loss in the spectral domain to enforce gaussianity. This is done by a loss that aligns the empirical power spectrum with its expected distribution i.e. $\chi_2$ distribution.
- Quantitative results on Aesthetic image generation and text-Aligned image generation show the effectiveness of the proposed regularizations in reward alignment.

**Questions:**

- In L314 authors claim that their method achieves higher rewards and outperforms baselines with fewer updates. Please provide experimental evidence to support this statement.
- Authors only focus on text to image models for their downstream task. I would recommend rewriting the paper to focus directly on that and propose the two regularizations instead of the way it is written right now for me to improve my ratings. In its current form, I find the paper very hard to follow.

**Ethical Concerns:**

["NO or VERY MINOR ethics concerns only"]

**Final Justification:**

Authors have addressed my concerns about using higher order moments in the matching loss. Authors also said they would make changes to the text which is hard to verify before camera ready. I trust the authors to make the necessary chanes to make the paper more readable and vote to increase my score.

**Limitations:**

yes.

**Quality:**

3

**Strengths And Weaknesses:**

**Strengths**
- The unification of existing methods under the spatial regularization proposed in this work is very informative and useful to the community.
- Qualitative and Quantitative results show the effectiveness of the proposed method.

**Weaknesses**
- The introduction and abstract was very hard to read. After reading the abstract, as a reader, I wasn't sure what was the application of the current method. Authors talk briefly about their method in the abstract but what is not clear to me was "Why is any of this necessary?".
Authors claim their method is useful for a range of downstream tasks but only provide results for reward alignment in text to image models. Then, a better way to organize the paper and the abstract would be to first state that for reward alignment, latents are optimized, and it is important to keep them close to the gaussian manifold. The paper would read very well if it is rewritten as a latent regularization loss for reward alignment instead of its current form.
- Authors claim gaussianity is important and propose a spatial regularization term but end up using 1st and 2nd moments only in the loss. Why not enforce more terms since matching more moments implies getting much closer to the gaussian manifold.

---

> ### Author Rebuttal · Authors · 2025-07-31
>
> Thank you for taking the time to carefully review our work and provide valuable feedback. We address the concerns below.
>
> ---
> ***Q:*** *After reading the abstract, as a reader, I wasn't sure what was the application of the current method.*; *The paper would read very well if it is rewritten as a latent regularization loss for reward alignment instead of its current form.*
>
> ***A:*** We appreciate your comment regarding the clarity of our paper. Our intention was to introduce the proposed regularization loss in a broader context, as Gaussianity is a fundamental concept with wide applicability across various domains. However, we understand that this broader framing may have made it difficult to identify the specific focus of our work. In response to your helpful suggestion, we will revise the paper to more clearly position it as a latent regularization method for reward alignment.
>
> ---
> ***Q:*** *Why not enforce more terms since matching more moments implies getting much closer to the gaussian manifold.*
>
> ***A:*** We report additional results using different $\mathcal{K}$ values in text-aligned image generation:
>
> | $\mathcal{K}$ | $\emptyset$| \{1\} | \{1,2\} | \{1,2,3,4\} | \{1,...,6\} | \{1,...,8\} |
> |---|---|---|---|---|---|---|
> | **PickScore**$\uparrow$ | 0.2484 | 0.2500 | 0.2548 | 0.2510 | 0.2521 | 0.2525 |
> | **HPSv2**$\uparrow$ | 0.2933 | 0.2962 | 0.3028 | 0.3012 | 0.2999 | 0.2984 |
>
> The performance with \{1,2\} shows a clear performance improvement over both $\emptyset$ and \{1\}. However, we did not observe further improvements when extending to higher-order moment losses. The first two terms were sufficient to surpass baseline performance, as our regularization loss was designed not only to incorporate moment-based losses but also to include a power-spectrum-based loss that effectively suppresses residual spatial structure. (**L42**, **L186**) Additionally, we provide a unified framework that integrates regularization losses which had previously been used independently. (**Section 3.1**)
>
> ---
> ***Q:*** *In L314 authors claim that their method achieves higher rewards and outperforms baselines with fewer updates. Please provide experimental evidence to support this statement.*
>
> ***A:*** We acknowledge that this expression may be misleading. Our intended meaning is that the proposed regularization helps to reach *comparable* reward with *fewer* optimization steps. We will revise the sentence to improve clarity as follows:
>
> “*As a result, it achieves higher rewards over baselines for the same iterations.*”
>
> ---

---

> > ### Author Response · Authors · 2025-08-07
> >
> > Thank you for your time and participation in the review process. As the discussion phase approaches its end, we would like to ask if you have any remaining questions or comments. We are prepared to address any further input you may wish to provide.

---

### Official Review · Reviewer_Thdj · 2025-07-07

**Clarity:** 2
**Significance:** 2
**Originality:** 2
**Rating:** 5
**Confidence:** 3

**Summary:**

The Gaussian distribution plays a central role in optimizing diffusion-based models. In particular, when optimizing diffusion for reward alignment, it is important to incorporate a regularization technique to minimize reward hacking and other negative effects of optimization. This paper investigates existing regularization techniques using the structural properties of the standard Gaussian distribution and proposes a novel regularization method that incorporates information from both spatial and spectral domains. The paper performs experiments on reward alignment in a one-step image generation model and text-aligned image generation. The results show that, compared to baselines such as no regularization, KL divergence, kurtosis, ReNO, and PRNO, the proposed regularization technique outperforms them all.

**Questions:**

- How sensitive is \mathcal{K} in the joint objective?
- Did you experiment with moments higher than 2?
- How sensitive are the experiments to \lambda_\text{power}?
- Is the joint objective just a combination of KL and PRNO? If so, can you elaborate on why regularization is needed in both the spectral and spatial domains simultaneously?
- Could you explain how spectral and spatial domain regularization work together?

**Ethical Concerns:**

["NO or VERY MINOR ethics concerns only"]

**Final Justification:**

The author addressed my concerns about specific conceptual details in the paper.

**Limitations:**

Yes

**Quality:**

3

**Strengths And Weaknesses:**

Strengths:

When performing reward alignment, it is crucial to regularize models to prevent reward hacking and degradation in performance. This paper introduces a novel, simplified formulation of alignment, synthesizing existing work by demonstrating that most current approaches are optimizing a subset of moments of a Gaussian distribution that have a general form, as presented. Not only is the paper well-written, but it also clarifies how the proposed idea relates to previous methods, making the unification clear.

Weakness:

The paper appears to lack clear ablation studies demonstrating how sensitive the $\mathcal{K}$ or $\lambda_\text{power}$ parameters are in practice. For example, presenting ablation experiments with different \mathcal{K} and \lambda_\text{power} values would offer valuable insight into which factor has the greatest impact. Additionally, since the observation indicates that KL matches the first two moments, it's important to clarify whether the primary contribution of the experiments is in adding the constraint within the spectrum domain, given \mathcal{K} is set to either 1 or 2.

---

> ### Author Rebuttal · Authors · 2025-07-31
>
> Thank you for your thoughtful review and valuable comments. We address the concerns in detail below.
>
> ---
> ***Q:*** *How sensitive is $\mathcal{K}$ in the joint objective?*; *Did you experiment with moments higher than 2?*
>
> ***A:*** We report additional results with different $\mathcal{K}$ values in text-aligned image generation:
>
> | $\mathcal{K}$ | $\emptyset$| \{1\} | \{1,2\} | \{1,2,3,4\} | \{1,...,6\} | \{1,...,8\} |
> |---|---|---|---|---|---|---|
> | **PickScore**$\uparrow$ | 0.2484 | 0.2500 | 0.2548 | 0.2510 | 0.2521 | 0.2525 |
> | **HPSv2**$\uparrow$ | 0.2933 | 0.2962 | 0.3028 | 0.3012 | 0.2999 | 0.2984 |
>
> The performance with \{1,2\} is clearly improved compared to $\emptyset$ and \{1\}. However, we did not observe further improvements when extending to higher-order moment losses. The first two terms were sufficient to surpass baseline performance, as our regularization loss was designed not only to incorporate moment-based losses but also to include a power-spectrum-based loss that effectively suppresses residual spatial structure. (**Section 3.2**) Additionally, we provide a unified framework that integrates regularization losses which had previously been used independently. (**Section 3.1**)
>
> ---
> ***Q:*** *How sensitive are the experiments to $\lambda_\text{power}$?*
>
> ***A:*** We present the following results on text-aligned image generation with varying $\lambda_{\text{power}}$:
>
> | $\lambda_{\text{power}}$| 0 | 5 | 10 | 25 | 50 | 100 |
> |---|---|---|---|---|---|---|
> | **PickScore**$\uparrow$ | 0.2481 | 0.2516 | 0.2513 | 0.2548 | 0.2521 | 0.2533 |
> | **HPSv2**$\uparrow$ | 0.2968 | 0.2974 | 0.3032 | 0.3028 | 0.3060 | 0.3042 |
>
> We observe that the performance remains stable across a wide range of $\lambda_{\text{power}}$ values, and is consistently better than the case with $\lambda_{\text{power}} = 0$. These results indicate that the method is **not sensitive** to this parameter.
>
> ---
> ***Q:*** *whether the primary contribution of the experiments is in adding the constraint within the spectrum domain*
>
> ***A:*** We would like to clarify that our contributions extend beyond simply introducing a spectral-domain constraint.
>
> - **Moment-based loss (Section 3.1):** Unifies independently developed regularization terms such as **KL divergence**, **norm**, and **kurtosis** losses within a single framework.
>
> - **Power-spectrum-based loss (Section 3.2):** Defined via the **DFT** in the spectral domain, it regularizes DFT coefficients to match their theoretical distribution. Unlike **PRNO**, which removes spatial correlation by aligning the empirical covariance matrix with the identity and incurs quadratic complexity, our method uses **FFT** for efficient computation with $O(D \log D)$ complexity and operates via a fundamentally different mechanism.
>
> - **Dual-domain regularization (Section 3.3):** We apply spatial and spectral losses **jointly**, leveraging their **complementary strengths**—the spatial loss ensures marginal distribution alignment, while the spectral loss enforces spatial neutrality. (**L116**)
>
> Therefore, we emphasize that our contribution is not limited to the addition of a spectral loss, but lies in the integration and synergy of both domains within a coherent and efficient framework.
>
> ---
> ***Q:*** *Is the joint objective just a combination of KL and PRNO?*
>
> ***A:*** Our loss is not simply a combination of KL and PRNO. We would like to clarify two key points:
>
> - The primary distinction lies between **PRNO** and our **power-spectrum-based loss**. PRNO suppresses spatial correlation by encouraging the empirical covariance matrix to approximate the identity matrix (**L80**), which incurs **quadratic complexity**. In contrast, our method leverages the **FFT** to compute **DFT coefficients** and regularizes them to match the theoretical distribution. This approach results in a significantly lower complexity ($O(D \log D)$) (**L48**). As shown in **Figure 3**, PRNO takes **14.1 seconds** for 100 iterations, whereas our method requires only **0.26 seconds**, achieving over **50× speed-up**—while more effectively matching the Gaussian distribution.
>
> - We also emphasize the **complementary roles** of spatial and spectral regularization, as discussed in **Section 3** and illustrated in **Figure 1**. While both KL and PRNO do not address regularization in the spectral domain, our formulation incorporates **spectral-domain regularization** to more effectively enforce Gaussianity. Importantly, our regularization loss shows **superior performance** compared to these baselines in the experiments.
>
> ---
> ***Q:*** *Why regularization is needed in both the spectral and spatial domains simultaneously*; *How spectral and spatial domain regularization work together*
>
> ***A:*** Moment-based losses **permutation-invariant** and therefore do not account for the **ordering** of elements in a sequence. (**L185**) For example, consider a set of 10K Gaussian samples that have been sorted. Although this sequence does not resemble a natural realization of random Gaussian samples—since such perfect ordering is highly improbable—the moment-based loss cannot distinguish it from a genuinely random Gaussian sample because it only matches marginal statistics.
>
> To address this, we propose a power-spectrum-based loss, which is **permutation-variant** and sensitive to spatial correlations. (**L43**, **L186**) By analyzing the DFT coefficients, it captures deviations from white Gaussian noise and penalizes residual spatial structure. In **Lemma 1**, we provide a theoretical characterization of the expected distribution of the power spectrum under the assumption of independently sampled Gaussian noise in the spatial domain. This formal foundation justifies the use of spectral loss as a principled way to enforce spatial neutrality.
>
> However, the **spectral loss alone** does not ensure alignment with the target **marginal distribution**, as it focuses only on spatial structure. Therefore, **both losses must be used together**:
>
> - The **moment-based loss** ensures the correct marginal distribution.
>
> - The **spectral loss** enforces the absence of spatial correlation.
>
> These two losses play **complementary roles** in fully regularizing toward Gaussianity (**L116**). To further support this, we provide visual evidence in **Figure 1**, which illustrates why regularization in both domains is necessary.
>
> ---

---

> > ### Author Response · Authors · 2025-08-07
> >
> > Thank you very much for your time and engagement throughout the review process. As we near the end of the discussion phase, we would greatly appreciate any remaining questions or comments you might wish to share. We welcome all further input and are eager to provide any clarifications or revisions necessary.

---

> > > ### Comment · Reviewer_Thdj · 2025-08-08
> > >
> > > Thank you for the response and addressing my concerns. I will raise my score.

---

> > > > ### Author Response · Authors · 2025-08-09
> > > >
> > > > We sincerely appreciate your acknowledgment of our rebuttal and your confirmation that our clarifications have addressed the concerns, as well as your decision to raise the final score.

---

### Decision · Program_Chairs · 2025-09-17

**Decision:**

Accept (poster)

**Comment:**

The paper introduces a novel regularization technique for text-to-image models that ensures Gaussianity in the latent space. This technique aims to prevent issues like "reward hacking" during reward alignment optimization. The method uses a dual-domain regularization approach, combining:
- Moment-based loss (spatial domain): This unifies existing regularization methods (KL divergence, norm, kurtosis) into a single framework to ensure correct marginal distribution.
- Power-spectrum-based loss (spectral domain): Defined via the Discrete Fourier Transform (DFT), this loss is sensitive to spatial correlations, enforcing spatial neutrality and penalizing residual spatial structure. It uses Fast Fourier Transform (FFT) for significantly faster computation compared to prior methods like PRNO. The paper claims that these two losses play complementary roles: the moment-based loss ensures the correct marginal distribution, while the spectral loss enforces the absence of spatial correlation.

(b) Strengths:
- The paper proposes a novel regularization method and provides a unified theoretical framework that encompasses existing Gaussianity regularization methods under a moment-matching paradigm.
- It tackles the crucial issue of reward hacking and performance degradation during reward alignment in diffusion models.
- The spectral domain regularization achieves significantly improved computational efficiency (over 50x speed-up compared to PRNO) due to its use of FFT.
- The proposed technique outperforms baselines such as no regularization, KL divergence, kurtosis, ReNO, and PRNO in experiments on reward alignment for one-step image generation and text-aligned image generation.
- The method is well-motivated and theoretically grounded, with a formal foundation for using spectral loss to enforce spatial neutrality.

(c) Weaknesses
- While theoretical foundations were provided, a deeper analysis of why dual-domain regularization is necessary was requested by reviewers
- The paper appeared to lack clear ablation studies demonstrating the sensitivity of the K (number of moments) and $\lambda$ power parameters.
• The authors acknowledged that the loss function value alone cannot reliably indicate how closely a latent vector matches a true standard Gaussian distribution, which was noted as a limitation.

(d) Reasons to accept: The paper proposes a novel and technically solid regularization method that effectively addresses the important problem of reward hacking in text-to-image models. Its unified framework and efficient dual-domain approach are innovative contributions. The authors provided compelling experimental results demonstrating superior performance and effectively addressed the reviewers' concerns during the rebuttal period.

(e) The authors effectively addressed reviewer concerns, leading to increased scores across the board.
- Reviewer Thdj asked about parameter sensitivity and the necessity of dual-domain regularization. Authors showed stability for $\lambda$ power and explained how moment-based and power-spectrum losses are complementary (spatial for marginal distribution, spectral for spatial neutrality/correlations), and clarified their method is not just KL+PRNO due to FFT efficiency (50x speed-up) and different mechanisms.
- Reviewer LvLi requested clarification on the paper's main application and whether more moments should be enforced. Authors agreed to rewrite for clearer focus on latent regularization for reward alignment and confirmed higher-order moments provided no further performance gain. They also clarified their "fewer updates" claim meant higher rewards for the same iterations.
- Reviewer p3aM sought deeper theoretical analysis for dual-domain and generalization. Authors provided further theoretical justification for dual-domain necessity (complementary limitations of permutation-invariant moment losses and permutation-variant spectral losses). They also presented new results on SDXL-Turbo, demonstrating generalization beyond the initially tested model.
- Reviewer mYPt also requested broader evaluation and more on higher-order moments. Authors provided SDXL-Turbo results, confirming generalization, and reiterated that matching moments beyond the first two did not yield improvements. They also showed that jointly optimizing text embeddings offered no performance benefit.